

# Documents, Reanalysis, and Global Circulation Models: A New Method for Reconstructing Historical Climate Focusing on Present-day Inland Tanzania, 1856–1890

Philip Gooding[1], Melissa J. Lazenby[2], Michael R. Frogley[2], Cecile Dai[1], and Wenqi Su[1]

[1] Indian Ocean World Centre, McGill University, Montreal, H3A 0E6, Canada

[2] Department of Geography, University of Sussex, Brighton, BN1 9RH, United Kingdom

*Correspondence to:* Philip Gooding (philip.gooding@mcgill.ca); Melissa J. Lazenby (m.lazenby@sussex.ac.uk)

**Abstract.** This article proposes a novel methodology for reconstructing past climatic conditions in regions and time-periods for which there is limited evidence from documentary and natural proxy sources. Focusing on present-day inland Tanzania during the period 1856–1890, it integrates evidence from qualitative documentary sources with quantitative outputs from climate reanalysis and global circulation models (GCMs), which enables the creation of interdisciplinary seasonal time-series of rainfall variability for three distinct locales. It does so by indexing each dataset to the same 7-point scale and weighting each output according to a predefined level of confidence in the documentary data. This process challenges the subjectivity of nineteenth-century Europeans in Africa, whose reports form the basis of the documentary material, and adds evidence from the region, which is currently lacking from the latest reanalysis products and GCMs. The result is a more scientifically grounded interpretation of documentary materials and a more locally grounded estimation of rainfall that would otherwise be gained from referring to reanalysis or GCMs alone. The methodology is validated with reference to observed long-term trends gathered from (paleo)limnological studies, and it is shown to provide marked insights into four periods of environmental stress in the region's late-nineteenth-century past. Future challenges may involve integrating evidence from oral traditions and adapting the methodology for other regions and time-periods.

## 1 Introduction

Building models that project the future effects of climate change is partly dependent on understanding climatic variability in the past. Since around the mid-twentieth century, this fact has acted as a mandate for historical climatologists to reconstruct past climatic conditions through the use of existing statistical records (e.g. rain gauges), natural proxies (e.g. tree rings), and climate forcings (e.g. well-mixed greenhouse gases), which have led to the creation of a number of global climate models, drought atlases, and reanalysis products. Additionally, in the last twenty years, climate historians have increasingly sought to make contributions, largely through indexing qualitative descriptions of weather and climate held in archives (Nash et al., 2021; Adamson et al., 2022). This article integrates these two methodologies to make interdisciplinary time-series for seasonal rainfall in three locations in what is now inland Tanzania during the second half of the nineteenth century.

Integration of climatological and historical methods for this case study is a necessity borne out of the nature and quality of the source material. While climate reanalysis and global circulation models (GCMs), including the Twentieth Century Reanalysis Project (20CRv3) and the Climate Modelling Intercomparison Project (CMIP5), estimate rainfall in the region for the period under review, they either incorporate very few proxies or no proxies at all (natural or archival) from



Africa. For Africanists in the humanities and social sciences, Africa's absence from the underlying data makes deploying these projections uncomfortable, as it partly represents the continent's marginalisation from global scientific discourse during and after colonialism. Nevertheless, the models are tantalising, especially to East Africanists. Depending on the product used, they suggest daily, monthly, and annual estimations of rainfall, all of which represent a greater temporal resolution than that which
is gatherable from the natural proxies in the region that have been analysed up to now, which are mostly limited to reconstructions of past lake levels (Nicholson and Yin, 2001; Verschuren et al., 2000; Bessems et al., 2008; Russell and Johnson, 2007; Alin and Cohen, 2003).

The historical documents for late-nineteenth-century inland Tanzania are additionally problematic (cf. Brönnimann and Wintzer, 2019). They were written almost entirely by Europeans, including by so-called 'explorers' and missionaries, who
regularly had distorted understandings of the peoples, environments, and climates they encountered (Endfield and Nash, 2002; Gooding, 2019; Gooding, 2022a; Rockel, 2022). Also, they discussed climatic conditions irregularly. For example, although the first Europeans to document enviro-climatic conditions in inland Tanzania did so in 1856–61, another did not do so again until 1869 (Burton 1860; Speke 1864a; Speke 1864b; Livingstone 1875). Further, no individual European documented changing climate in a particular locale until the first missionaries settled in Mpwapwa in 1876, and even then, their reporting
was uneven. Thus, any indexed time-series made solely from European documents would necessarily rely on thin data, especially for the earlier decades under review, for which there is often no data at all.

Therefore, this article seeks to create time-series of seasonal rainfall that incorporates data from both global circulation models/reanalysis and documentary sources. It argues that evaluating the climatological materials against documents for seasons/years where the documents provide ample evidence allows them to be deployed with a certain degree of (un)certainty
in years where documentary data is lacking. Where the documentary data are thin, the models improve the precision and/or revise conclusions that may otherwise have been made; where the documentary data are non-existent, the models can be used in lieu, with a qualified amount of (un)certainty. In making these claims, the article also provides a rubric for historians and climatologists to use each other's sources to improve their reconstructions of past climatic conditions in different locales. This methodology provides climatologists with a more robust picture of past climates with added value from documentary datasets.
Likewise, it enables historians to make scientifically grounded interpretations of documentary materials that refer to climatic phenomena.

The case study focuses on three towns in present-day inland Tanzania: Mpwapwa, Tabora, and Ujiji (Kigoma) (Figure 1). These locales are chosen for several reasons. First, they were the three most important inland towns on a long-distance caravan route between the East African coast and Lake Tanganyika during the nineteenth century, on which ivory was the
primary article of commerce. They thus have historical significance (Rockel, 2006; Pallaver, 2020; Gooding, 2022b; Castryck, 2022). Second, and partly because of their significance, nineteenth-century Europeans passed through and settled in their locales with more regularity than in other parts of present-day inland Tanzania, meaning that the documentary source base that refers to them is relatively abundant. Third, they are all on roughly the same latitude (between 4.5 °S and 6.5 °S) and experience rainy seasons at roughly the same time: Mpwapwa receives most of its rainfall between December and April; Tabora and Ujiji
between November and April. Thus, it is expected that they would receive abundant/normal/deficient levels of rainfall at roughly the same time – a phenomenon which is supported by existing climatological studies (Nicholson, 2017a).

To be clear, this is not the first interdisciplinary project to attempt to reconcile historical and climatological sources for equatorial eastern Africa. Sharon E. Nicholson was at the forefront of such efforts in the 1990s and 2000s, culminating in her masterful annual (Jan–Dec) reconstruction of Africa's rainfall variability in the nineteenth century, co-authored with Amin
K. Dezfuli and Douglas Klotter (Nicholson et al., 2012; Nicholson et al., 2018). This dataset was developed from



Figure 1. Map showing locations of 3 case study regions, Mpwapwa, Tabora, and Ujiji in inland Tanzania. The base map is taken from Wessel and Smith (1996). The shapefile for the green area marked as Tanzania is taken from Tanzania National Bureau of Statistics / UN OCHA ROSA (https://data.humdata.org/dataset/cod-ab-tza).






a mixture of rain gauge records, documentary sources, oral traditions, hydrological sources, and spatial reconstructions, although rain gauges are generally lacking for inland Tanzania. Further, Nicholson has since made calls for historians to improve the documentary source base of such reconstructions (Nicholson 2017b), and she has noted the difficulty of placing oral traditions in time. Additionally, the annual (Jan–Dec) timescale of Nicholson, Dezfuli, and Klotter's (2012) reconstruction

obscures seasonal variation in inland Tanzania's rainfall, as the region's rainy season transcends the beginning and end of a Gregorian year. Nevertheless, the hydrological sources, which are largely limited limnological studies, point to broad trends that must be borne in mind when creating any time-series of rainfall in the region, notably that they suggest a generally wet period from c. 1840 until the mid-late 1870s, and regular droughts from then until the early twentieth century (Hastenrath, 2001; Nicholson and Yin, 2001; Nicholson 1999).

This article, therefore, adds new documentary and climatological sources to the latest reconstruction of nineteenth-century African climate, focusing on a particular region's seasonality rather than being constrained by the Gregorian calendar, while also bearing in mind observed, long-term trends in the region. It is hoped that the methodology developed in this article, which provides a rubric for integrating historical and climatological data and suggests ways that global climate reconstructions may be used as a representation for rainfall when there is little or no documentary record, may be adapted for other regions

and time periods, including pre-1850 East Africa, for which oral traditions are essential to East Africanist historical writing.

The remainder of the article is divided into four sections. The first analyses the documentary data and the time-series developed from it, incorporating degrees of (un)certainty for different data points. The second analyses the global climate models/reanalysis and the trends for inland Tanzania's rainfall that can be observed from them. The third develops a methodology for integrating the historical and climatological data, and the final section describes the resultant time-series and

its implications for understanding rainfall patterns in inland Tanzania during the second half of the nineteenth century.

## 2 Documentary Sources: 'Explorers' and Missionaries

The documentary data referring to inland Tanzania in the years 1856–90 comes in two forms. For the first circa twenty years, it is comprised of 'explorer' sources, such as those written by Richard F. Burton (1860), John Hanning Speke (1864a; 1864b),

David Livingstone (1875), Henry Morton Stanley (1872; 1878), and Verney Lovett Cameron (1877). These Europeans visited the region at different times, often observing climatic and environmental phenomena as they travelled, and gathering information about previous seasons/years from informants. The second type of document are the letters and diaries of Europeans who resided in the region from c. 1876, who were mostly missionaries. They were representatives the Church Missionary Society (CMS), the White Fathers (Père Blancs), the London Missionary Society (LMS), and the International

African Association (Association Internationale Africaine, AIA). These sources provide a longer and more detailed time-series of data about climatic and environmental conditions. Thus, generally speaking, the precision and accuracy of the documentary data is greater for the final circa fifteen years of the period under review than for the first twenty, notwithstanding some exceptions.

Although each document-type provides somewhat distinct challenges for creating an indexed time-series of rainfall,

there are some prevailing themes. In general, 'explorers' and missionaries were highly interested in documenting climatic conditions and variations, although their interest varied over time and space, and between each author. Despite this interest being useful to historical climatologists and climate historians working in the present, their reasons being so interested are rooted in highly problematic discourses. Nineteenth-century meteorology, like cartography (Wisnicki 2008), was part of a wider practice that sought to impose European science, and thus Europeans' ideas of 'civilisation,' on East Africa(ns), erasing



indigenous patterns of human-environment interaction and their understandings of climate and weather. In addition, missionaries' regular reports of drought and associated hardships acted partly as justification for European intervention in African affairs, which they (wrongly) assumed would increase drought resiliency (Gooding, 2023; Kjekshus, 1996; Doyle, 2006). As has been argued elsewhere, these reports may have been exaggerated to provoke emotional responses from readerships at home, who funded their missions (Endfield and Nash, 2002). At other times, missionaries may have minimised

the degree of hardship to emphasise their missions' feasibility (Gooding, 2022a). In short, the documents are highly subjective, and they both affected and were influenced by imperial knowledge-making, contributing to the 'Scramble for Africa' from the mid-1880s.

Given this historical background, it is probably unsurprising that Europeans commented on climatic conditions more when they were extreme, such as in instances of severe drought or floods, than during months/seasons/years of regular rainfall.

Droughts and floods had adverse impacts on Europeans' and surrounding societies' everyday lives, and so were deemed worthy of reporting. Such reports also appealed to their readership's understanding of African climates and environments. But the reports still often lacked specificity. Europeans might instead refer to the health of crops and their ability to travel. For example, crops 'threatening to dry out before they were mature' is a strong indicator of drought in the months leading up to the statement, while a missionary report of having to divert his direction of travel because of a 'terribly flooded country' is probably indicative of excessive rainfall (A.G.M.Afr. Diaire de Bukumbi, 24 May 1884; CMS C/A6/O/16 Mackay to Smith-MacKenzie & Co, 16

May 1878). Thus, inferences must often be made from climate-affected phenomena, such as harvests, to suggest rainfall conditions. It also means that more certainty can be gathered from the documentary archive for extreme weather events than for years of average or close-to-average rainfall. Absence of discussion about climatic conditions may be indicative of regular rainfall, although such an assumption necessarily comes with a degree of uncertainty.

Data from 'explorer' sources additionally provides distinct challenges further to the fact that there are several temporal gaps between reports. Lack of knowledge about the region's climate may have led to errors in reporting. For example, Richard Burton, who in 1857–8 was a member of the first European party to travel to Lake Tanganyika, wrote that Ujiji's rainy season lasted from September to May, contrary to current scientific knowledge, which places the rainy season between November and April (Burton, 1860). There are at least two possible reasons for this discrepancy: First, that the rainy season was

particularly long in 1857–8, and that he made assumptions based on his own experience; or second, that he mistakenly integrated secondary information about the 'lake regions' of eastern Africa into his assessment of Ujiji's climate, as a September–May rainy season broadly aligns with conditions on the northern shores of Lake Victoria in present-day Uganda, notwithstanding that December-February is usually drier than September-November and March-May in the latter place. Nevertheless, a tantalising reference to a flooded river in a normally arid zone about 100 kilometres west of Mpwapwa in

September 1857 suggests that inland-Tanzania's rainy season may have begun significantly early in 1857–8 (Burton, 1860). At the same time, Burton (1860) wrote that the rainy season began around Tabora in November 1857, lasting until mid-May 1858. Unfortunately, given the time-gaps between different 'explorers' travels, there are no other documentary sources with which to verify or challenge Burton's reports. Thus, references such as these often provoke as many questions as answers, even though there might be a suggestion (in this instance) of an especially protracted rainy season in certain locales.

Other challenges arise from the missionary sources. The peak of their reports on climatic conditions occurred as they entered the region in the mid-late 1870s. This was likely informed by an enthusiasm for their 'civilising' project, of which imposition of European scientific methods and measurements was an important component. But references to climatic conditions and their effects often became more fleeting over time: other factors, such as relations with indigenous rulers, the successes/failures of gaining converts, and internal disputes, started to dominate missionary reporting from the early 1880s.



Additionally, adversity with populations in Ujiji led members of the LMS to abandon their station there in 1883, meaning that their reporting on climate over the town becomes less certain thereafter. Often, reports from nearby stations, such as at Kibanga and Kavala Island, are necessary to supplement the data. Meanwhile, in the same year, CMS missionaries temporarily moved the core of their Mpwapwa station 8 kilometres out of town to Kisokwe, on the edge of a perennial river, enabling them to irrigate their garden. Thus, some subsequent reports of productive garden work and abundant harvests are probably more

indicative of successful use of infrastructure than of rainfall (CMS G/3/A/5/O Cole to Lang, 17 Jan. 1883; CMS G/3/A/5/O Cole to Lang, 15 Feb. 1883; CMS G/3/A/5/O Cole to Lang, 1 July 1883).

    Notwithstanding these challenges, this article uses a 7-point index system to quantify the qualitative descriptions of rainfall variability and its effects. This makes it interoperable with Nicholson, Dezfuli, and Klotter's (2012) dataset for Africa's nineteenth-century rainfall, which also uses a seven-point system, thus allowing incremental improvements to an existing

dataset to be made, rather than seeking to replace it. The definitions of each index value are displayed in table 1. Additionally, each line of data is graded on a two-point scale indicating how 'certain' it is. A grade of 1 indicates a high degree of uncertainty, caused by ambiguity in the description, lack of verifying documentary material, the use of data from nearby locations as a proxy, and/or use of proxies that are imperfect indicators of rainfall conditions (such as abundant harvests from irrigated fields). Thus, almost all datapoints provided by 'explorers' are given an uncertainty grade of 1. A grade of 2 indicates a higher degree

of certainty, caused by the climate conditions being described in detail, by more than one source, and/or from within the locale to which the time-series applies (Mpwapwa, Tabora, or Ujiji). A visualisation of the data is shown in figures 2–4. Occasionally the data are abundant enough that changes within a season are observable; more frequently, however, an indexed value is attributed to the season as a whole.

Table 1. Definitions of index values used to interpret qualitative descriptions of rainfall variability and its effects contained within documentary materials.

| Value | Description |
|---|---|
| -3 | Drought-driven famine (cf. Rockel 2022) |
| -2 | Deficient rainfall |
| -1 | Below-average rainfall |
| 0 | Average rainfall |
| 1 | Above-average rainfall |
| 2 | Excessive rainfall |
| 3 | Rainfall-driven floods |






Figures 2–4. Time-series plot of the 3 locales and their indexed rainfall for the period 1856–1890 (Mpwapwa: 2; Tabora: 3; Ujiji: 4). Black solid bars indicate the highest level of certainty from archival data and grey bars indicate a reduced certainty level. Values of +0.05 = 0 on the index. They are indicated on the plots as +0.05 to distinguish them from the seasons in with no documentary data


Figure 2.

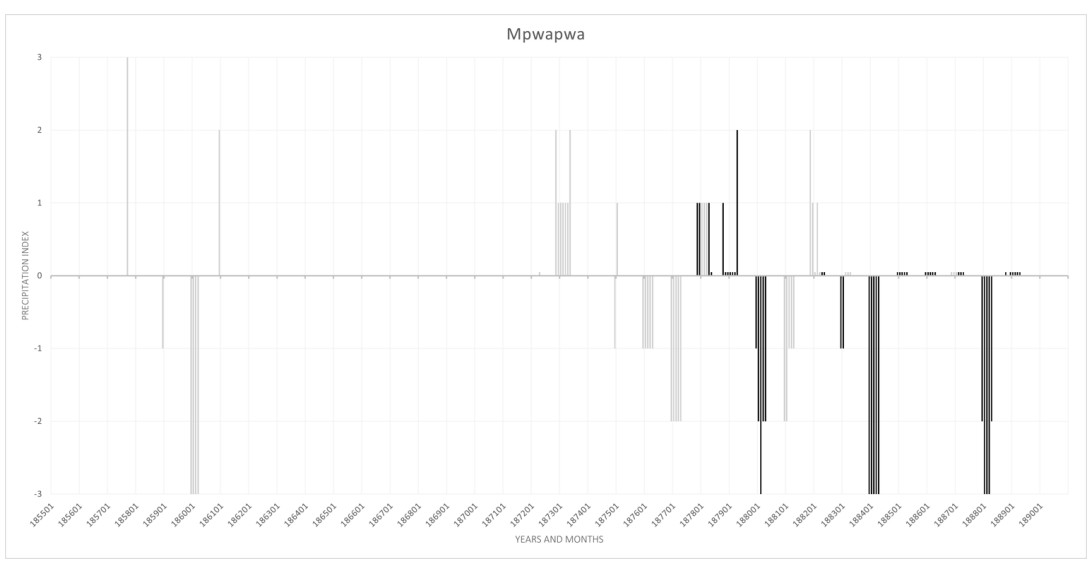

Figure 3.

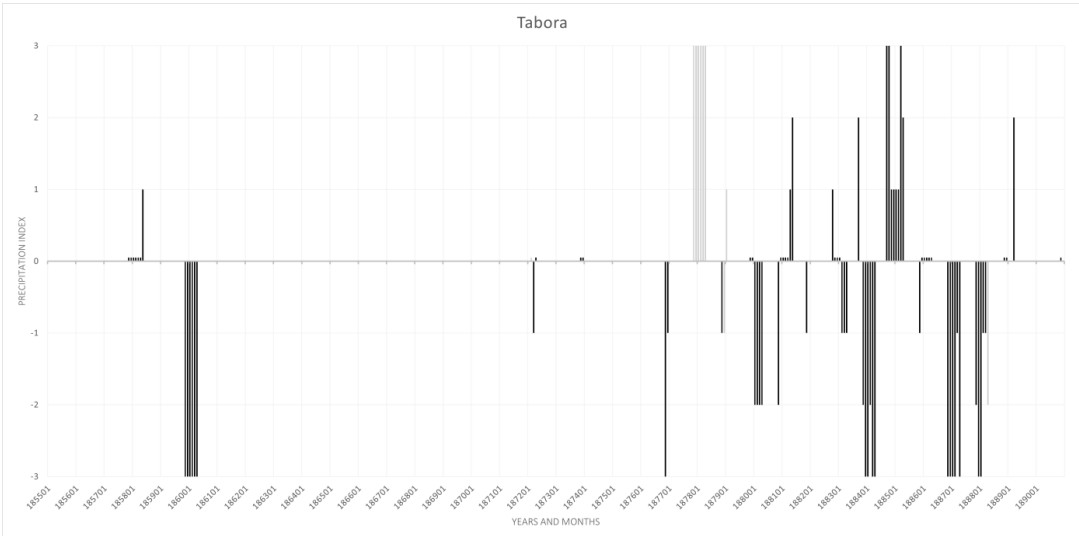




Figure 4.

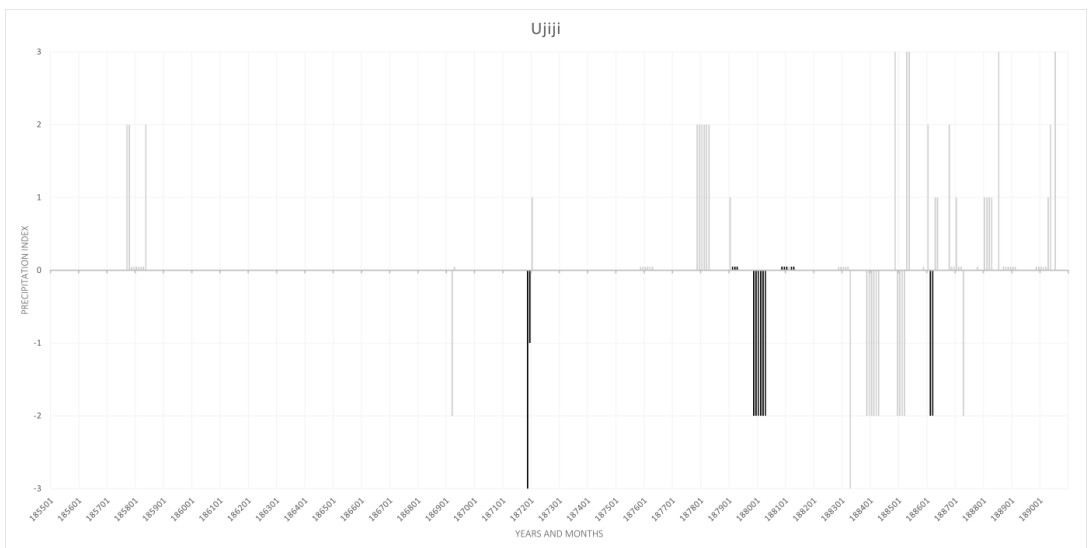

The three time-series reflect a total of 151 lines of data: 55 for Mpwapwa; 52 for Tabora; and 44 for Ujiji. Each line

refers to climate conditions during a given period, ranging from one month to an entire rainy season. The number of documentary references for each line is between one (several) and twelve (line 31 of the Ujiji dataset, which refers to November-December 1886, and includes data from a letter written by an LMS missionary based at Kavala Island and eleven separate diary entries by a White Fathers' missionary based at Kibanga). In addition to published data by 'explorers,' which provide references for all locales, data for Mpwapwa is informed principally by documents held in the CMS archive, with

documents from the White Fathers' archive providing additional information for 1880-82; the Tabora dataset is equally informed by documents in the CMS and White Fathers' archives, with occasional references to the LMS archive; and the Ujiji dataset is informed equally by references in the LMS and White Fathers' archives, as well as by one reference from the AIA archive. The datasets underpinning figures 2–4, including transcriptions and comments on individual references, are available by link in the section, 'Data Availability'.

In their totality, the documentary datasets indicate certain region-wide trends. As expected, the data are thin – in quantity and quality – for the first twenty years under review, but there is a marked improvement on both counts from the second half of the 1870s. In line with expectations from Nicholson, Dezfuli, and Klotter's 2012 dataset, drought seasons and years were regular from this point onwards, apart from during a possible flood event in 1877–78. These trends are largely supported by (paleo-)limnological research that was conducted in the 1990s–2000s, which indicates declining lake levels

across all the region's lakes from the late 1870s, a phenomenon that has regularly been attributed to frequent droughts (Nicholson and Yin, 2001). Additionally, the documents suggest severe drought across the region in the 1876–77, 1879–80, and 1883–84 seasons (although the sources are uncertain for Ujiji in 1883–4). As some historians have recently suggested, these climatic conditions may have been triggered by extreme El Niño Southern Oscillation (ENSO) and Indian Ocean Dipole (IOD) anomalies and atmospheric disturbances following the eruption of the Krakatau volcano in 1883 (Gooding, 2019;

Rockel, 2022; Gooding, 2022b; Gooding, 2022c). Further climatological research would help to verify or challenge these summations.



Additionally, there are some regional differences within the data. Although the sources are uncertain, the documents suggest regular levels of rainfall in Mpwapwa during the second half of the 1882–3 season, despite evidence for below-average rainfall around Tabora and Ujiji. This scenario at from Mpwapwa is made furthermore uncertain because the subsequent season

coincided with widespread famine. This may be indicative of previous harvests being insufficient, triggered by deficient levels of rainfall, decreasing societal resilience to region-wide drought in 1883–4, and so putting into question the validity of observations gathered from missionary correspondence in 1882–3.

Other differences between datasets are more immediately acceptable, however. The sources are explicit that rainfall was somewhat regular around Tabora and Mpwapwa in 1885–6, and that the region around Ujiji was comparatively dry in the

second half of the rainy season (although rainfall may have been excessive earlier on). For example, a member of the White Fathers who travelled between Tabora and Ujiji in early 1886, wrote that, having had abundant rainfall around Tabora up to February, "Almost everywhere we encountered nothing but perfectly dry roads, and more than once we saw ourselves on the point of suffering from thirst in places where usually, at such a season, one travels in water" (A.G.M.Afr. Unknown to White Fathers, 14 Mar. 1886). According to other regional sources, this drought affecting Ujiji may have extended to the southern

shores of Lake Victoria (A.G.M.Afr. Diaire de Bukumbi, 30 Dec. 1885, 2 Jan. 1886, 5 Jan. 1886, 15 Feb. 1886, 17 Feb. 1886, 23 Feb. 1886, 26 Feb. 1886, 4 Mar 1886, 11 Mar 1886, 1 May 1886, 2 May 1886).

Similarly, the sources indicate that the drought that affected Ujiji and Tabora in 1886–7 did not extend to Mpwapwa, and that the drought of 1887–8 affected Mpwapwa and Tabora, but not Ujiji. A diary entry in the White Fathers' archive, for example, directly contrasts an abundance of water around Mpwapwa with dryness in the vicinity of Tabora following the

1886–7 rainy season (A.G.M.Afr. Diaire de Kipalapala, 30 May 1887). Meanwhile, a rain gauge temporarily stationed on Kavala Island, Lake Tanganyika suggests regular rainfall in the first half of 1888 in Ujiji, contrary to several reports of drought in Tabora and Mpwapwa. The rain gauge further suggests that rain was more abundant in 1887–8 than in 1885–6 and 1886–7, two years for which the documents suggest there was drought (Hore, 1892). Meanwhile, in both 1886–7 and 1887–8, documentary sources suggest that drought also affected regions around Lake Victoria, including around present-day Kampala,

Uganda (A.G.M.Afr. Diaire de Bukumbi, 7 Nov. 1886, 14 Nov. 1886, 14 Dec. 1886, 17 Dec. 1886, 1 Jan. 1887, 13–15 Jan. 1887, 5 Feb. 1887, 6–7 Feb. 1887, 21 Feb. 1887, 23–25 Feb. 1887, 16 Dec. 1887, 11 Jan. 1888; A.G.M.Afr. Diaire de Rubaga, 21 Sep 1886, 9 Feb. 1887; A.G.M.Afr. Livinhac to Lavigerie, 15 Feb. 1887; A.G.M.Afr. C.14–535 Benoit to White Fathers, 28 June 1888; CMS G/3/A/5/O Mackay to Lang, 27 Dec. 1886). The documentary sources suggest that these were widespread droughts, perhaps associated with the aftereffects of Krakatau's eruption, although the geographical spread was uneven.


### 3 Global Climate Reconstructions (Reanalysis and GCMs)

The climatic reconstructions employed in this article come in two forms: climate reanalysis and global circulation models (GCMs). Climate reanalysis products use observational data from regions and time-periods where it is abundant to model climatic conditions in regions and time-periods where it is lacking, such as in inland Tanzania during the nineteenth century.

Specifically, we used the NOAA–CIRES–DOE Twentieth Century Reanalysis version 3 (20CRv3), which assimilates surface pressure observations to estimate climatic conditions, including precipitation rate (*prate*), over 1°x1° grid squares at a monthly scale from 1836–2015 (Slivinski et al., 2019). We also considered incorporating data from version 2 of the Last Millennium Reanalysis (LMR) and the Paleo Hydrodynamics Data Assimilation (PHYDA) product (Tardif et al., 2019; Steiger et al., 2018). However, both these reanalysis products operate at an annual timescale (LMR: Jan–Dec; PHYDA: Apr–Mar), and the

PHYDA reanalysis only provides outputs that are suggestive of *prate* (for example, Palmer Drought Severity Index [PDSI]





and Standardized Precipitation Evapotranspiration Index [SPEI]). Thus, lack of interoperability of these data with the seasonal data being collected hindered their incorporation into the interdisciplinary datasets being constructed here.

Unlike climate reanalysis, GCMs do not rely on observational or paleoclimatic data. Rather, they simulate the climate system based on natural (orbital, solar, volcanic) and anthropogenic (well-mixed greenhouse gases, ozone, tropospheric aerosols, land use) climate forcings. Here, we employed historical twentieth-century model simulations from a total of 25 models from the World Climate Research Programme (WCRP) Coupled Model Intercomparison Project Phase 5 (CMIP5) multi-model dataset, in the Fifth Assessment Report of the Intergovernmental Panel on Climate Change (Meehl et al., 2007; Taylor et al., 2012). All model data were re-gridded to 1.5°x1.5° to ensure uniformity. Model simulations from the Paleomodel Intercomparison Project (PMIP) were also explored, but unavailability of significant quantities of data from after 1850 limited their utility (Otto-Bliesner et al., 2009, Schmidt et al., 2012). By contrast, several time-series from the CMIP5 model runs begin in 1850. These data were obtained from the Program for Climate Model Diagnosis and Inter-comparison (PCMDI). The first ensemble members (r1i1p1) of CMIP5 historical runs were extracted, but only for models that had data beginning from 1850 (see Table 2). The reanalysis and GCM datasets are visualised in figures 5–10 in terms of anomalies in mm/day for prate and precipitation respectively.


Table 2. List of the 25 CMIP5 models used.

| Model Name | Modeling Center (or Group) | Institute ID | Atmospheric Resolution |
|---|---|---|---|
| ACCESS1.0<br><br>ACCESS1.3 | Commonwealth Scientific and Industrial Research Organization (CSIRO) and Bureau of Meteorology (BOM), Australia | CSIRO-BOM | 1.25˚ x 1.9˚ |
| BCC-CSM1.1 | Beijing Climate Center, China Meteorological Administration | BCC | 2.8˚ x 2.8˚ |
| BNU-ESM | College of Global Change and Earth System Science, Beijing Normal University | GCESS | 2.8˚ x 2.8˚ |
| CanESM2 | Canadian Centre for Climate Modeling and Analysis | CCCMA | 2.8˚ x 2.8˚ |
| CCSM4 | National Center for Atmospheric Research | NCAR | 0.94˚ x 1.25˚ |
| CESM1(BGC)<br><br>CESM1(CAM5) | Community Earth System Model Contributors | NSF-DOE-NCAR | 0.94˚ x 1.25˚ |
| CMCC-CM<br><br>CMCC-CMS | Centro Euro-Mediterraneo per I Cambiamenti Climatici | CMCC | 0.75˚ x 0.75˚<br><br>1.9˚ x 1.9˚ |
| CNRM-CM5 | Centre National de Recherches Météorologiques / Centre Européen de Recherche et Formation Avancée | CNRM-CERFACS | 1.4˚ x 1.4˚ |



| | | | |
|---|---|---|---|
| | en Calcul Scientifique | | |
| **CSIRO-Mk3.6.0** | Commonwealth Scientific and Industrial Research Organization in collaboration with Queensland Climate Change Centre of Excellence | CSIRO-QCCCE | 1.9˚ x 1.9˚ |
| **EC-EARTH** | EC-EARTH consortium | EC-EARTH | 1.1˚ x 1.1˚ |
| **FIO-ESM** | The First Institute of Oceanography, SOA, China | FIO | 2.8˚ x 2.8˚ |
| **INM-CM4** | Institute for Numerical Mathematics | INM | 1.5˚ x 2˚ |
| **IPSL-CM5A-LR** <br><br> **IPSL-CM5A-MR** <br><br> **IPSL-CM5B-LR** | Institut Pierre-Simon Laplace | IPSL | 1.9˚ x 3.75˚ <br><br> 1.25˚ x 2.5˚ <br><br> 1.9˚ x 3.75˚ |
| **MIROC-ESM** | Japan Agency for Marine-Earth Science and Technology, Atmosphere and Ocean Research Institute (The University of Tokyo), and National Institute for Environmental Studies | MIROC | 2.8˚ x 2.8˚ |
| **MIROC5** | Atmosphere and Ocean Research Institute (The University of Tokyo), National Institute for Environmental Studies, and Japan Agency for Marine-Earth Science and Technology | MIROC | 1.4˚ x 1.4˚ |
| **MPI-ESM-LR** <br><br> **MPI-ESM-MR** | Max-Planck-Institutfür Meteorologie (Max Planck Institute for Meteorology) | MPI-M | 1.9˚ x 1.9˚ |
| **MRI-CGCM3** | Meteorological Research Institute | MRI | 1.1˚ x 1.1˚ |
| **NorESM1-M** <br><br> **NorESM1-ME** | Norwegian Climate Centre | NCC | 1.9˚ x 2.5˚ |





Figures 5, 7, and 9. Time-series showing the 20CRv3 precipitation rate anomalies for Mpwapwa for the season DJFMA (5),
Tabora for the season NDJFMA (7), and Ujiji for the season NDJFMA (9). Blue bars indicate positive precipitation rate
anomalies and red bars negative precipitation rate anomalies.

Figures 6, 8, and 10. Time-series showing the 25 multimodel mean CMIP5 precipitation anomalies for Mpwapwa for the
season DJFMA (6), Tabora for the season NDJFMA (8), and Ujiji for the season NDJFMA (10). Blue bars indicate positive
precipitation anomalies and red bars negative precipitation anomalies.


Figure 5.

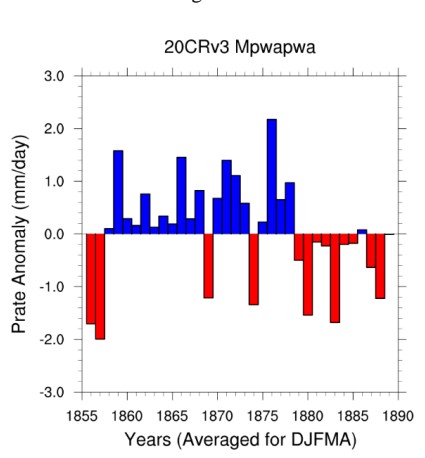

Figure 6.

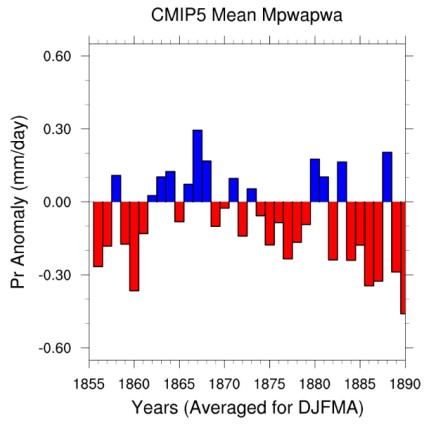

Figure 7.

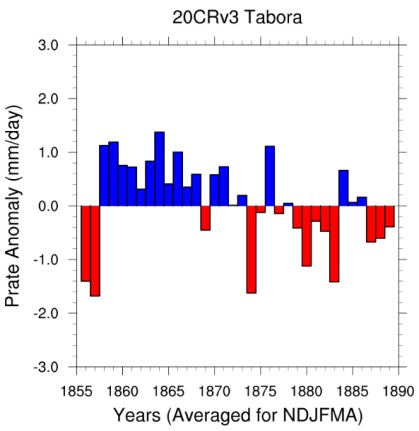

Figure 8.

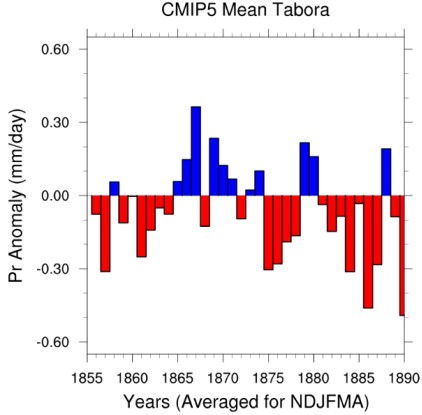



Figure 9.                                             Figure 10.

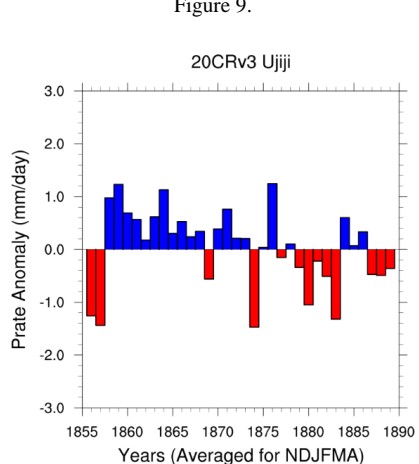    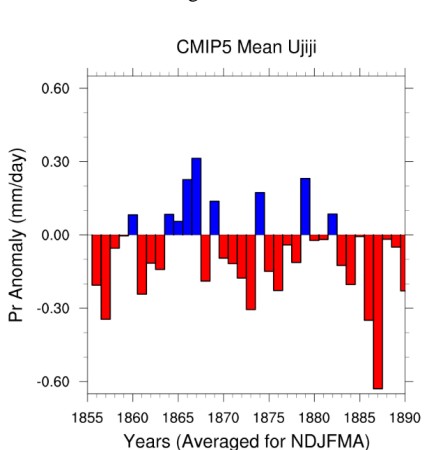

There are marked discrepancies between the outputs for each locale, some of which are immediately explainable. For example, the scale of rainfall anomalies varies widely between the reanalysis and GCM datasets. This is at least partly attributable to the fact that the outputs from the CMIP5 datasets are multi-model ensemble means, indicating that output anomalies are typically small due to some offsetting between different models. Thus, large anomalies, which are occasionally apparent in the 20CRv3 and indexed documentary datasets have been obscured, as is further evident from the much more amplified anomalies indicated in individual runs (see figure 11 for individual CMIP5 runs for Mpwapwa, as an example).

For the first twenty years of the period under review, the 20CRv3 outputs suggests generally average or above-average rainfall across almost all seasons. This is in line with most hydrological sources from the region (Nicholson and Yin, 2001). The CMIP5 datasets on the other hand, intersperse positive and negative anomalies with a predominance of the former, suggesting several years of abundant rainfall around the late 1860s and early 1870s, as well as deficient rainfall during the mid-1870s. This is broadly in line with some evidence from the indexed dataset, as well as from reports of a slight decline in the level of Lake Tanganyika around this time, before increasing again around 1877–8 (Nicholson 1999). Indeed, the negative precipitation anomalies are stronger and more frequent for Ujiji than for Mpwapwa and Tabora, suggesting that there may have been lower levels of rainfall in Lake Tanganyika's catchment than in other parts of present-day inland Tanzania.

Despite these discrepancies, there are two major areas of agreement between most of the climatological datasets. First, there appears to have been a significant region-wide drought towards the end of the 1850s, although its exact timing and duration is unclear. Incorporating data from the documentary dataset, which relies on data contained in John Hanning Speke's account, will add temporal precision in this context. Even so, the fact the climatological sources suggest that the drought affected Ujiji indicates that it extended to regions far beyond where Speke travelled. Second, drought is universally indicated in the mid-late 1880s, especially in the 1883–4 season. This is notwithstanding the occasional year of relatively regular rainfall, as suggested by the 20CRv3 reanalysis. Generally speaking, and in line with hydrological sources from the region and relevant parts of the documentary datasets, the reanalysis and GCMs indicate that the period c. 1860–75 was wetter than the period c. 1875–1890.

Figure 11. CMIP5 panel plot for 25 individual models indicating Mpwapwa's rainfall anomalies for DJFMA from 1856-57–1889-90. Model titles are embedded within each individual plot (see overleaf).





**4 Integrating Documentary and Climatological Data**

Integration of documentary, reanalysis, and GCM datasets to make an interdisciplinary, seasonal time-series for each locale necessitated beginning with a two-step process for making each dataset interoperable. First, we modified the temporal resolution of the indexed documentary datasets so that each season could be expressed using a single numerical value. It is acknowledged that this process could potentially obscure in-season rainfall variability that is apparent in some documentary accounts from the 1870s–80s. For example, the highly variable rainfall suggested in the missionary archive for Ujiji in 1885–

86 ranged between 'excessive' (Jan), 'deficient' (Feb–Mar), and 'above average' (Apr), but probably amounted to regular levels of rainfall in the totality for the season (see figure 4). Nevertheless, our approach gave a strong indication of overall rainfall levels across entire seasons and brought the temporal resolution of the indexed documentary dataset into line with those of the reanalysis and GCM datasets, facilitating intercomparison at the same temporal scales. Figures 12–14 indicate seasonal rainfall according to the indexed documentary dataset, as well as levels of confidence in each.


Figures 12–14. Seasonal Documentary Precipitation Index Values for Mpwapwa (12), Tabora (13), and Ujiji (14). Red bars indicate a confidence level (CL) of 2 and pink indicate a confidence level of 1. Values of +0.05 = 0 on the index. They are indicated on the plots as +0.05 to distinguish them from the seasons in with no documentary data.

Figure 12.                                                                  Figure 13.

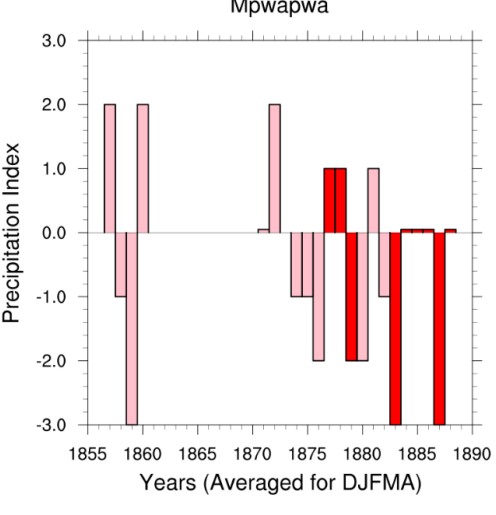
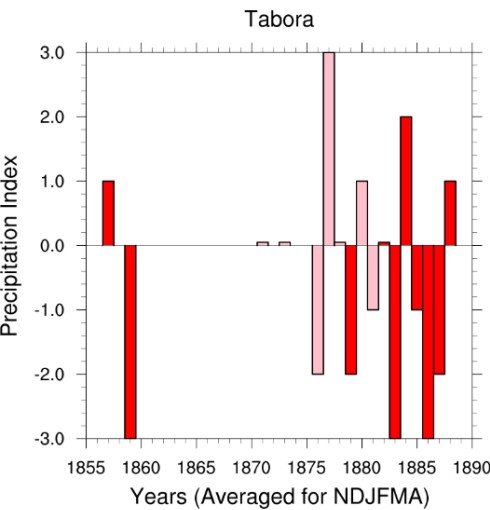





Figure 14.

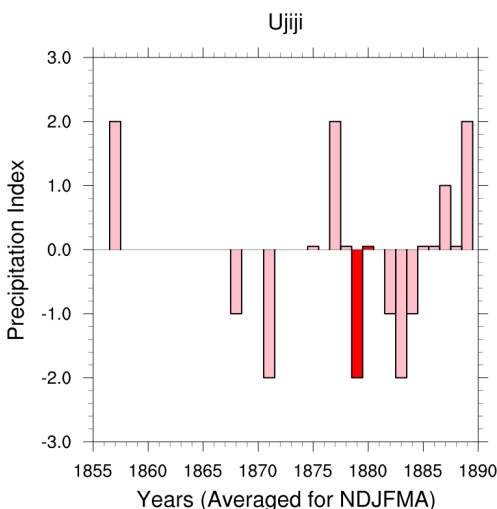

The second step required interpreting the reanalysis and GCM datapoints in terms of indexed anomalies, rather than in terms of millimetres of precipitation per day (mm/day). The outputs for each dataset were converted into a seven-point precipitation scale, in line with the indexed documentary datasets. This required multiplying each reanalysis and GCM dataset

by a factor that would make the most extreme anomaly equal to +/–3.5. For the reanalysis, this was a factor of 1.6 (the most extreme value was +2.2 for 1876-77 in Mpwapwa); for the GCM, this was a factor of 5.5 (most extreme value was –0.63 for 1886–87 in Ujiji). The assumption was made that all indexed reanalysis and GCM values between –3.5 and –2.5 were broadly representative of the definitions applied for values of –3 in the indexed documentary dataset, that reanalysis and GCM values between –2.5 and –1.5 were indicative of indexed documentary values of –2, and so on. This approach therefore allowed

conversion of the reanalysis and GCM data to the same scale as the documentary data, enabling direct comparisons and integration, whilst also maintaining the temporal resolution of both reanalysis and GCM datasets. The three indexed datasets for each locale are combined and visualised in Figures 15–17.

Figures 15 (Mpwapwa) and 16 (Tabora) suggest similar patterns of rainfall variability up to 1860 and then an overall a wetting trend (across all three datasets) up to around 1873. The subsequent interval is then characterised by a broad

dominance of drying anomalies until the end of the period of interest. Ujiji displays the most dataset variability of all three localities (figure 17), though also has the least documentary data available. Broad trends nevertheless suggest that wetter conditions in the first half of the period are followed by drier conditions, broadly in agreement with the other two localities. It is also clear from all datasets that the 1883–84 season was anomalously dry at every locality.

In integrating the datasets, divergent rules were used depending on the availability and confidence level in each

datapoint from the documentary data index. When the confidence level value in the documentary datapoint was 1, equal weight was given to the documentary, reanalysis, and GCM datasets (33.3%/33.3%/33.3%). This was justified on the basis that the documentary datapoints often lacked specificity and/or verifying contextual material, and occasionally relied on information from outside the locality under review. In this sense, it suffered from some of the same constraints of the reanalysis and GCM




Figures 15–17. Time series for Mpwapwa (15), Tabora (16), and Ujiji (17) including all 3 datasets which have been extrapolated to create a uniform precipitation index (Pr Index) ranging from -3.5 to +3.5. CL 1 in the documentary datasets is shown in pink and CL2 in red.

365                                                    Figure 15.

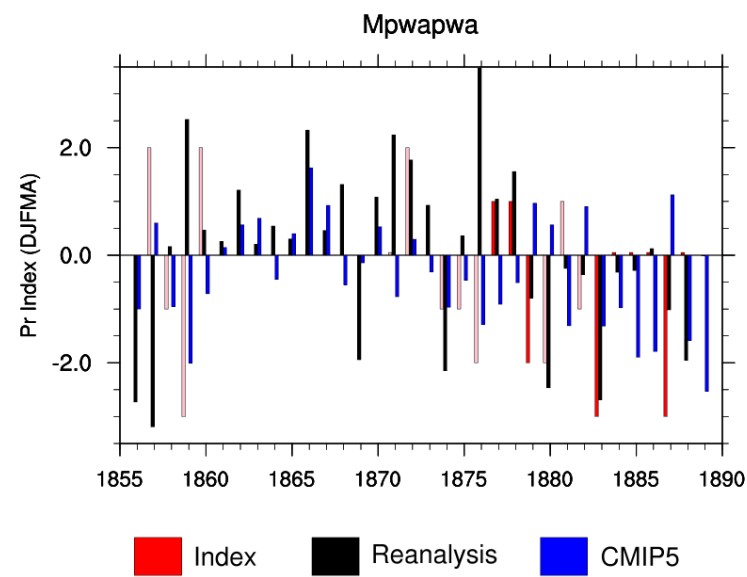


Figure 16.

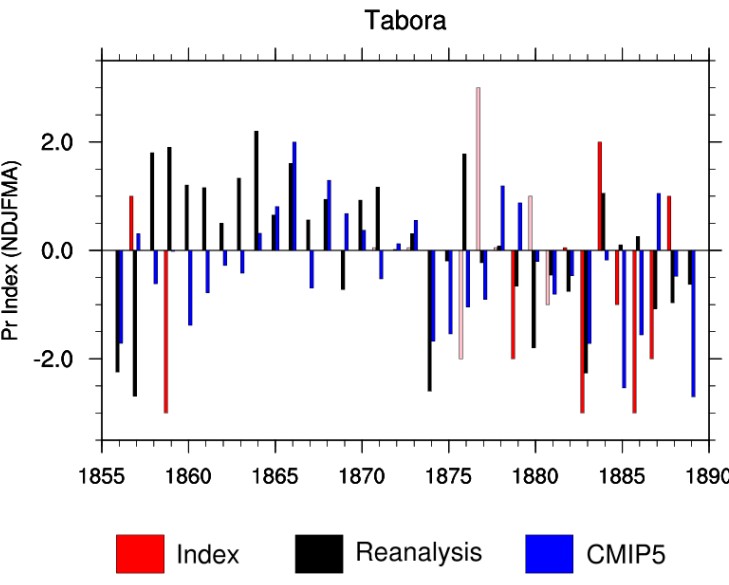





Figure 17.

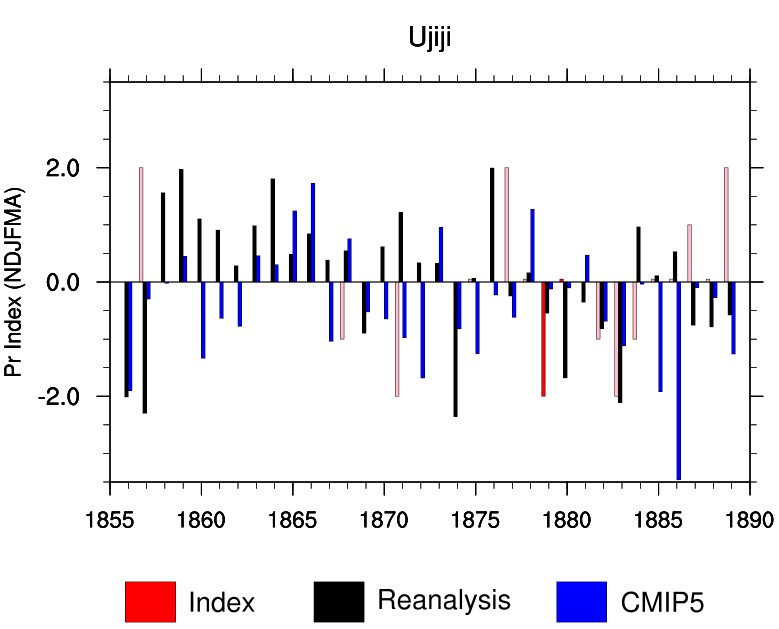

datapoints, which also rely on data collected over a broader geographical region. By contrast, given that documentary
datapoints with a confidence level of 2 are the only ones to regularly incorporate cross-verified data from within each of the
three localities, more weight was applied to these when integrating them with the reanalysis and GCM datasets
(50%/25%/25%). When documentary data were absent, equal weight was given to the reanalysis and GCM values (50%/50%).
Correlations were calculated using both unweighted and weighted values (see tables 3 and 4). The outputs in the resultant time-
series were then multiplied by a factor of 1.4 to minimise offsetting of extremes during the integration process (figure 18).

Table 3: Time-series correlations between datasets for all 3 localities for the period 1856–57 to 1889–90 using Pearson's
correlation, including missing values. All significant correlations at the 95th percentile level are indicated in bold

| Correlations (1856–57 to 1889–90) | Mpwapwa | Mpwapwa Weighted | Tabora | Tabora Weighted | Ujiji | Ujiji Weighted |
|---|---|---|---|---|---|---|
| Index Data vs 20CR | 0.09 | 0.10 | –0.05 | 0.01 | –0.13 | –0.08 |
| Index Data vs CMIP5 | 0.03 | 0.00 | 0.10 | 0.05 | –0.03 | –0.03 |
| 20CR vs CMIP5 | –0.01 | 0.12 | 0.24 | **0.36** | 0.22 | 0.28 |



The correlations between the datasets are low, which is expected due to the relative shortness of the time-series and the missing values in the indexed documentary datasets. It is worth noting, however, that higher correlations are evident

between the reanalysis dataset and CMIP5 model dataset for Tabora and Ujiji. This provides higher confidence that the overall trends in these localities are likely to be more robust, as the two climatological datasets show some coherence. None of the above correlations are statistically significant at the 95th percentile except for the weighted 20CR versus the CMIP5 dataset. Table 4 is similar to Table 3, with the main difference being that missing values were excluded when calculating the correlations. None of the correlations were significant, most likely due to the reduced number of datapoints overall.


Table 4: Time-series correlations between datasets for all 3 localities for the period 1856–57 to 1889–90 using Pearson's correlation, excluding missing values. There are no significant correlations at the 95th percentile level.

| Correlations (1856–57 to 1889–90) | Mpwapwa | Mpwapwa Weighted | Tabora | Tabora Weighted | Ujiji | Ujiji Weighted |
|---|---|---|---|---|---|---|
| Index Data vs 20CR | 0.08 | 0.10 | –0.15 | –0.13 | –0.27 | –0.26 |
| Index Data vs CMIP5 | –0.04 | –0.07 | 0.16 | 0.15 | –0.06 | –0.08 |
| 20CR vs CMIP5 | –0.29 | –0.38 | –0.08 | –0.10 | –0.10 | –0.09 |

Additional correlations were carried out that separately incorporated data at confidence level values of either 2

(highest) or 1. One significant correlation emerged between the Index Data and 20CR dataset for Mpwapwa of $r = 0.73$ at the confidence level value of 2. This is promising, as it suggests that higher confidence in the indexed documentary data improves correlation with the reanalysis dataset, which is as close to observations as possible given the time-period and location. All other correlations were insignificant at the 95th percentile level but are included in table 5 for reference.

Table 5. Time-series correlations between datasets for all 3 localities for the period 1856–57 to 1889–90 using Pearson's correlation, excluding missing values for different Index Data Confidence Levels (CL). All significant correlations at the 95th percentile level are in bold.

| Correlations (1856/57–1889/90) | Mpwapwa CL 1 | Mpwapwa CL 2 | Tabora CL 1 | Tabora CL 2 | Ujiji CL 1 | Ujiji CL 2 |
|---|---|---|---|---|---|---|
| Index Data vs 20CR | –0.22 | **0.73** | –0.52 | –0.12 | –0.32 | *n/a |
| Index Data vs CMIP5 | 0.35 | –0.53 | 0.04 | 0.17 | –0.02 | n/a |
| 20CR vs CMIP5 | –0.55 | 0.02 | –0.18 | –0.08 | –0.05 | n/a |

*n/a: there were only 2 datapoints at confidence level 2 for Ujiji which makes it insufficient to obtain a correlation coefficient value.





Figure 18. Time-series representing Integrated and Weighted Precipitation Anomalies for All Localities.

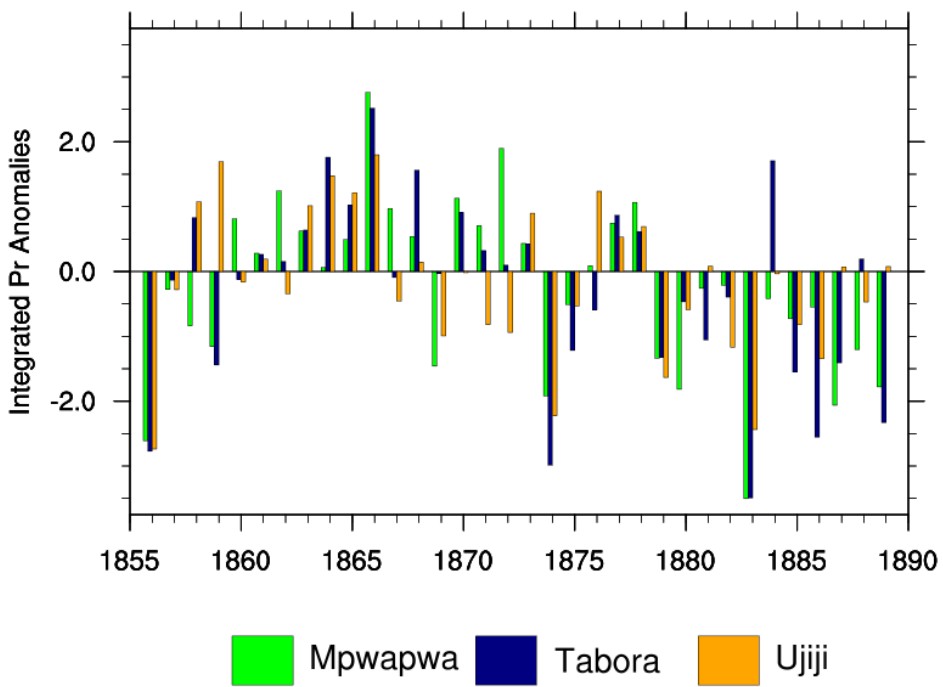

Figure 18 suggests that there are several outputs in common across each of the 3 localities with large positive or negative precipitation anomalies. The most notable of these are 1856–57, 1866–67, 1874–75, 1879–80 and 1883–84. Of the 34 years analysed in total, the integrated dataset shows that 20 (or 59%) of those years agree in the sign of the precipitation anomaly (i.e., conditions becoming wetter or drier). In quantifying the degree of statistical correlation between the three localities (again using Pearson's correlation coefficients), the relationship is strongest between Mpwapwa and Tabora (r = 0.75), then Tabora and Ujiji (r = 0.68) and lastly Mpwapwa and Ujiji (r = 0.54). All three correlations are significant at the 95th percentile level using a student t-test. This implies that the trends suggested by the newly integrated dataset were experienced similarly across all three localities. Overall, this outcome is broadly what might be expected due to their geographical proximity and similar seasonal climatology (Nicholson 2017a).

**5 Implications of the Resultant Time-Series**

The time-series represented in figure 18 have several implications for how historians and climatologists can interpret rainfall patterns in inland Tanzania during the second half of the nineteenth century. From a long-term perspective, they reinforce observed trends from (palaeo-)limnological studies that were incorporated into Nicholson, Dezfuli, and Klotter's (2012) dataset. They suggest a generally wet period during the third quarter of the nineteenth century, followed by a period of regular and severe droughts. The one exception to this is the record for Ujiji, which suggests slightly below average rainfall for most of the 1870s. This may reflect depressed levels of rainfall in the catchment of Lake Tanganyika during the first part of the





decade, as a limnological study suggests its level decreased slightly during this period, before peaking again during a region-
wide season of excessive rainfall in c. 1877–78 (Nicholson 1999). On a more granular level, however, the time-series improve
on the precision of existing climatological studies and standard interpretations of archival materials referring to rainfall. Four
examples are discussed in turn below.

### 5.1 1856–60 Droughts

Existing climatological studies do not point to drought in inland Tanzania during this period. Nicholson, Dezfuli, and Klotter
(2012) suggest consecutive seasons of regular and/or slightly above average rainfall. Also, whilst historian Juhani Koponen
(1988) used evidence derived from oral traditions to suggest a period of widespread drought during the 1860s, our time-series
suggest that an earlier date for such a phenomenon may be more appropriate. This may be in line with some interpretations of
traditions evidence from Ugogo (the region directly west of Mpwapwa) that date a famine called *Chonya-Magulu* (literally:
"hobble") to the 1850s or c. 1860 (Rockel, 1997). At the same time, standard readings of Speke's account of his second visit
to the region would place the drought, which contributed to famine around Tabora and towards Mpwapwa, only in 1859–60
season (Speke, 1864a). However, the time-series suggest that the famine probably resulted instead from a longer-term period
of environmental adversity that reached its apex as Speke visited. Drought may have been putting pressure on agricultural
production for several years leading up to 1859–60, decreasing reserves that may otherwise have mitigated against crop failure.
Moreover, warfare (which Speke (1864a) reported on) and migration may have further hindered the 1859–60 harvest, while
also increasing demand for food, especially around Tabora (cf. McDow, 2018; Sheriff, 1987). Below average or deficient
levels of rainfall in 1859–60 probably contributed to a longer-term environmental crisis affecting parts of inland Central
Tanzania in the late 1850s.

Nevertheless, there is still some uncertainty about the geographical extent of this drought. Although both the
reanalysis and GCM datasets indicate severe drought across the region in 1857–58, this contrasts with the inference from
Burton's account that the rainy season started early in Ujiji during that year, owing to his claim that Ujiji's rainy season
regularly began in September. The climatological data suggests that Burton and the natural inference from his account is
probably incorrect. However, both the reanalysis and GCM datasets also suggest abundant rainfall in 1859–60, but this is not
supported by the much more certain account provided by Speke in these years. In this instance, Speke's reports unequivocally
indicate drought around Mpwapwa and Tabora, and so the interdisciplinary time-series revise down the estimations of rainfall
made by the reanalysis and GCM datasets for these years. The challenge, however, is that Speke's account of this season does
not provide any data for Ujiji, and so the estimation for rainfall in that locale may be erroneously high, relying as it does only
on the reanalysis and GCM time-series. Here, only the incorporation of higher resolution natural proxies could provide more
certainty about the depth, duration, and spatial extent of the drought.

### 5.2 1876–78 Drought and Floods

Climatic conditions in equatorial eastern Africa in 1876–78 have recently garnered historical attention in the context of wider
anomalies affecting the Indian Ocean monsoon zone, which were triggered by extreme positive El Niño and Indian Ocean
Dipole events (Gooding, 2019; Gooding, 2022a; Singh et al., 2018). This research, which relied on missionary sources,
suggested widespread drought in the 1876–77 season and abundant rainfall in 1877–78 season. The time-series support this
hypothesis to a point. Although they support the general pattern (at least in Tabora), the level of anomaly may not have been



as extreme as previously suggested. In this instance, the fact that missionaries were only just entering the region in 1876 may have influenced them to overstate the unusual nature of the climatic conditions they witnessed, partly because they were yet to understand what was and was not normal. However, these years are also among those with the greatest divergence between datasets that underpin the time-series. In 1876–77, the reanalysis dataset suggests excessive levels of rainfall across the region, and in 1877–78, the GCM suggests below average levels of rainfall. Such results are inconsistent with the usual teleconnections between East African climate and positive and ENSO and IOD events – although these relationships are still not fully understood (Nicholson, 2015). Thus, inconsistency in the climate models/reanalysis may have offset extreme anomalies that are suggested in the documentary data, and which are reflected in the wider region affected by the Indian Ocean monsoon.

**5.3 1883–84 Drought**

The time-series suggest that the 1883–84 season experienced the deepest drought during the period under review. This is in line with recent historical work that emphasises the importance of rainfall deficiency in this season to a range of historical processes, including food security, epidemics, trade, migration, and political stability, which in turn may have undermined resistance to the imposition of colonial rule in the 1890s (Rockel, 2022; Gooding, 2022c; Gooding, 2023). It also challenges Nicholson, Dezfuli, and Klotter's (2012) characterisation of the season, which suggests that conditions in 1883–84 were much like they were throughout most the 1880s. The reason that this season stands out in the time-series is that all three datasets – the documents, reanalysis, and GCM – agree that rainfall was at least "below average" in this year, with most points suggesting either "deficient" or "famine" levels. In this sense, the climatological materials support standard interpretations of the archive, giving scientific support for recent historical research.

**5.4 1885–88 Droughts**

The time-series point to below average or deficient levels of rainfall across most of the region in three consecutive rainy seasons between the end of 1885 and April 1888. This is broadly in line with Nicholson, Dezfuli, and Klotter's (2012) dataset, in that they suggest below average or deficient levels of rainfall across the years 1886–88, with 1887 being the year of the deepest and most widespread drought. It does, however, contravene standard interpretations of the documentary material, which suggest that drought conditions did not become established until the 1886–87 season in Tabora, the 1887–88 season in Mpwapwa, or at all in Ujiji. Thus, absence of discussion about drought in missionary correspondence in 1885–86 may reflect the fact that a dearth of rainfall had yet to severely impact the societies in which they lived, or that rainfall levels were high relative to recent seasons, such as in 1883–84. Missionaries then only began to report on it when drought-related shortages began to affect them in subsequent years of deficient rainfall, especially in Mpwapwa and Tabora, for which the documentary record is relatively abundant. Meanwhile, incorporation of climatological materials for Ujiji suggests that the deficient levels of rainfall witnessed in February–March 1887 began earlier in the season, and that the "uncertain" reports of excessive rainfall (see figure 4) in January may be inaccurate.

**5.5 (Inter-)Disciplinary Implications**

From a historian's perspective, incorporation of climatological data allows for a more scientifically grounded method for analysing documentary materials that refer to climatic variability and its effects. This is especially important for the case



studies highlighted in this article, for which the documentary material is thin in both quantity and quality. Thus, reports of drought, which are regular in the archive for present-day inland Tanzania over the period 1856–90, can be contextualised within longer term climatic trends. A recurring theme is that such reports may often be more indicative of when droughts began to adversely affect the societies within which the Europeans lived or visited, not when the drought set in. Thus, incorporation of climatological materials challenges the subjectivity of nineteenth-century European observers in the region under review.

It is expected that this approach could be used in other regions, especially in the Global South, where European-authored texts dominate the documentary record, even though those same European authors regularly misunderstood the climatic and environmental contexts they reported on.

From a climatologist's standpoint, meanwhile, integration of documentary sources with reanalysis and GCM data adds much-needed precision to existing knowledge about past climatic conditions in inland Tanzania. Such a methodology is particularly necessary because high-resolution natural proxies have yet to be investigated in the region. Moreover, the ways in which global climatic teleconnections affect levels of rainfall in this region are still relatively poorly understood (Nicholson 2017a). Thus, direct evidence from the region in the form of historical documents, hitherto overlooked by historical climatologists focusing on inland East African climate, allow the models to be deployed with greater certainty and with an interpretation that is at least somewhat locally grounded, notwithstanding some discrepancies between historical and climatological data. Confidence in the reanalysis and GCMs is supported by a general correlation between the levels of their estimations of rainfall and what might be expected from analysis of (paleo-)limnological research in the region. This allows them to be deployed with a qualified degree of (un)certainty in the seasons for which documentary data are absent.

## 6 Conclusion

The methods that underpin this article, which incorporate and integrate qualitative documentary data and quantitative data from climate reanalysis and GCMs, are replicable for other regions and time-periods. They will be especially valuable for case studies for which the documentary and/or natural proxy data lacks resolution, quantity, and/or quality. The results of using such methods, however, can also be refined, as knowledge about the drivers of global and regional climate variability improves. A challenge moving forwards will be to establish how incorporation of documentary materials into reanalysis models can be achieved at a wider temporal and spatial scale. Achieving this will add precision to current reanalysis models, which will further improve the precision of interdisciplinary time-series, such as those provided with this dataset.

For inland Tanzania and surrounding regions, the next challenge will be to suggest rainfall variability for periods before the beginning of the documentary era. The 20CRv3 dataset may assist in this context, at least as far back as 1836, before which data are still under construction. Other reanalysis datasets, such as the LMR and PHYDA products, may be deployed for earlier periods, although the outputs would have a lower temporal resolution. Using a more estimative approach may also be compatible with historians' usage of oral traditions, which, up to now, they have had difficulty placing in time. Only with the analysis of high-resolution natural proxies in the region, however, could seasonal rainfall variation be reliably estimated for periods before and including the early nineteenth century.



**Data Availability**

The datasets that underpin the indexed time-series made solely from documentary data (figures 2–4) and the integrated indexed time-series (figure 18) are archived in the McGill University Dataverse, and are freely available under a CC-BY license at
https://doi.org/10.5683/SP3/LDODGI.

**Author Contributions**

Philip Gooding proposed initial ideas, analysed documentary data, co-developed project methods, and co-wrote the manuscript. Melissa Lazenby analysed reanalysis and GCM data, produced the figures, co-developed project methods, and co-wrote the
manuscript. Mick Frogley co-developed project methods and edited the manuscript. Cecile Dai and Wenqi Su performed initial experiments and analyses to ensure the feasibility of the project.

**Competing Interests**

The authors declare that they have no conflict of interest.


**Acknowledgements**

The research for this article was funded by the Social Sciences and Humanities Research Council of Canada. The authors wish to thank Daniele Battistelli and Riccardo Mercatali for their collection of documentary materials from the White Fathers' Archive. Part of this research was first presented at Australian National University's *Environmental Exchanges* seminar series
on 19 April 2023, and the authors would like to thank all those in attendance for their feedback and questions. The authors also acknowledge the World Climate Research Programme's Working Group on Coupled Modelling responsible for the CMIP5 model data, which were provided by the Program for Climate Model Diagnosis and Intercomparison (PCMDI). More information on these model data can be found at the PCMDI website (https://esgf-node.llnl.gov/projects/esgf-llnl/). Additionally, the authors acknowledge the U.S. Department of Energy, Office of Science Biological and Environmental
Research (BER), the National Oceanic and Atmospheric Administration Climate Program Office, and the NOAA Physical Sciences Laboratory, which supports the Twentieth Century Reanalysis Project version 3 dataset. More information can be found at the Physical Sciences Laboratory Website (https://psl.noaa.gov/data/20thC_Rean/).

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
