# Peer review of "Documents, Reanalysis, and Global Circulation Models: A New Method for Reconstructing Historical Climate Focusing on Present-day Inland Tanzania, 1856–1890"

_EGUsphere, 2023_

## Author Comment (AC1)

Dear Prof. David Nash,

Thank you very much for your kind, considered, and very constructive comments. We have made several changes based on your queries and recommendations. Apart from including them in a revised draft, we have detailed our responses below.

Best,

The Authors

**General comments**

This is a welcome paper addressing the thorny issue of how best to integrate historical and meteorological evidence to reconstruct climates of the past. It is clearly structured, generally very well written and makes persuasive arguments. However, as you will see from my specific comments below, it includes some statements that are problematic. These mainly relate to the way that the text falls into the "documentary evidence unreliable, instrumental data reliable" trope common to many climatological studies.

> Agreed. We have made several edits to revise the tone of the work so that the documentary evidence is given more of its due. Our intention was certainly not to diminish the utility and importance of the documentary-derived data – our time-series rely significantly on it, after all. Notable edits are on lines: 51-2, 120-4, 180-2.

There are also some methodological issues that need addressing. These include the need for a clearer description of the way in which documentary index classes are derived.

> Descriptions added to Table 1 (line 197).

Explanation is also needed as to why – given that the goal of the study is to produce a time series that is "interoperable with Nicholson et al. (2012)" (line 167) – the method used to convert modelled rainfall levels from 20CR and GCMs is not the same as that used by Nicholson et al. (2012).

> This error stems from Philip Gooding (Author 1) misinterpreting what 'interoperable' means – apologies. Now that he has informed himself, we can state that the aim was not to make the dataset interoperable with Nicholson et al. (2012). Instead, we think our outputs, which use a 7-point scale, may be

"broadly comparable" (edit made) with the Nicholson et al. (2012) dataset because they use the same scale, even if they are not entirely interoperable because we use different sources and methods. The choice to use the 7-point scale was made because the data was in many places granular and detailed enough that we could be this precise (instead of limiting ourselves to a 3-point of 5-point scale). Again, edit made to reflect this. We think this edit also helps to respond to the reviewer's first point (lines: 177-82)

I'm very surprised that the authors do not cite the recent papers by Nash et al. (2018) and Mutua & Runguma (2020), which present 19th century documentary climate series for Malawi and Kenya respectively. I would have thought that these are essential for comparison with the results presented for Tanzania.

Thank you for these references. We've read them and made a number of changes that incorporates evidence from them. We've also added evidence from rain gauges in Kampala (1879, 1881-1886), Mombasa (1875-1881), and Zanzibar (1874-1881), which are referred to in the data for Nicholson et al. (2012). These help to situate the findings with data already gathered from the wider region. See esp. lines: 93-5, 123-4, 469-79.

Mutua, T.M. and Runguma, S.N. (2020) Documentary driven chronologies of rainfall variability for Kenya, 1845–1976, Journal of Climatology and Weather Forecasting, 8, 255, available at: https://www.longdom.org/open-access/documentary-drivenchronologies-of-rainfall-variability-for-kenya–18451976.pdf

Nash, D.J. et al. (2018) Rainfall variability over Malawi during the late 19th century, International Journal of Climatology, 38 (Suppl. 1), e629–e642.

**Specific comments**

Lines 16-17, 60 and 507 – I'm not sure the wording "…a more scientifically grounded interpretation of documentary materials…" in the abstract and main text is ideal. This makes a value judgement about the validity of documentary evidence. Maybe 'climatologically grounded' rather than 'more scientifically' grounded would be better.

This is a good point. We have changed it to, 'an interpretation of documentary materials that is grounded in both the humanities and natural sciences.' See lines: 16-17, 63-4, 560.

Line 38 (and throughout) – best to avoid the use of terms derived from 'East Africa' as this is a colonial construct. Academics that I have worked with from the region tend to prefer the term 'eastern Africa'.

> This is true, and something Philip Gooding (Author 1) addressed in his recent monograph (*On the Frontiers of the Indian Ocean World: A History of Lake Tanganyika, c.1830-1890*, Cambridge, 2022, p. xii). Unfortunately, 'eastern Africa' doesn't really work either, as the term can be (and has been) applied to anywhere in the eastern half of the continent, from Cairo in the North to the Cape in the South. Thus, it lacks specificity. The better term is 'equatorial eastern Africa,' but this is perhaps overly long and its usage may hinder readability. We also noted that several climatological studies (e.g. Alin and Cohen 2003; Bessems et al., 2008; Hastenrath 2001; Nicholson 2015; Nicholson and Yin 2001; Verschuren et al. 2000) use 'East Africa', and so we thought we were conforming to established practice. But, in the interest of challenging colonial constructs, perhaps this practice needs to change. Thus, we have changed all references to East Africa to equatorial eastern Africa. However, if there are objections related to readability, we would rather revert to East Africa (which is usually understood to refer to Tanzania, Kenya, Uganda, Burundi, Rwanda, and the eastern DRC) than to adopt 'eastern Africa,' because of the latter's lack of specificity.

Line 45 – I don't think Endfield & Nash (2002) used a term as strong as 'distorted' to describe European perceptions (and hence descriptions) of African climate. Rather, the descriptions made by Europeans were often framed relative to their 'home' climate (particularly during their early years of residence in Africa), so tend to over-emphasise drier conditions. As noted in section 2, they may also be shaped by imperial knowledge-making.

- We've clarified this. True – their understandings of climate on its own were not necessarily 'distorted.' However, the role that they saw climate playing in the regions they reported certainly was i.e. they conceptualisation was rooted in enviro-climatic determinism. We've made an edit to reflect this and taken out the reference to Endfield and Nash 2002, which does not deal with this issue as much as the other cited texts. See lines: 46-7.

Lines 46-48 – this is a very strong statement – are you sure that absolutely no records made by Europeans describing climate in Tanzania exist between 1861 and 1868? I find that very hard to believe.

> This is actually correct, although we've made an edit to clarify. No Europeans entered the region between Speke/Grant's expedition (which ended in 1861) and

David Livingstone's arrival in Ujiji in 1869. The clarification we've made is to point out that there were no first-hand reports between these dates (this is similar to inland Kenya, as Matua and Runguma (2020) point out as well). There may have been the odd second-hand report made by administrators in Zanzibar, who may have consulted with Omani/African traders and/or porters who had returned from inland regions, but we have not made an effort to find them, if they even exist. If they do, they will not provide the kind of granular data that the firsthand reports that we have consulted contain. Line 49.

Lines 114-127 – this paragraph makes some valuable points. However, it paints the rather sweeping picture that all explorer and missionary descriptions of weather and climate were shaped by imperial agendas and are therefore unreliable. Some descriptions might well be 'highly subjective' – especially the broad overviews of climatic conditions in the more general explorer monographs – but other accounts of specific weather events and related phenomena (e.g. delays to the start of the rainy season, counts of rainy days, descriptions of flood events, descriptions of pasture conditions etc) will likely be reliable. I suggest that this paragraph be tweaked to provide greater nuance.

> Yes, the paragraph, as it stood, skewed overly negative. Thus, we have added 'positive' aspects of the documentary material as well: "Descriptions of rainy days, flood events, pasture conditions, and harvests, which are all to varying degrees related to climatic conditions, are abundant and can be used to make time-series of rainfall variability (Nash et al., 2018; Matua and Runguma 2020)." (Lines 123-4). We've kept the references to imperial knowledge-making, however. We feel it important to acknowledge how and why these documents exist, as well their utility to scholars in the present (the latter being much more well-known).

Lines 128-129 – the emphasis on describing extreme conditions is not unique to African documentary evidence and is well documented in historical climatology studies around the world – have a look at some of the excellent reviews by Christian Pfister or Rudolf Brazdil for further details and cite relevant methodological sources.

> Thank you. Added references to Pfister 1995, Brazdil 2000, and Brazdil et al. 2005, as well as a call out to this wider research in the opening line of the paragraph: "Given this historical background (and in line with other documentary sources for other regions and time periods), it is probably unsurprising that Europeans commented on climatic conditions more when they were extreme, such as in instances of severe drought or floods, than during months/seasons/years of regular rainfall." Line: 135-8.

Lines 130-139 – these kinds of uncertainty surrounding 'climatically indirect' indicators of climate variability are routinely dealt with in historical climatology studies and there is a wide literature on this. Again, have a look at some of the reviews by Christian Pfister or Rudolf Brazdil for details. These include explicit guidance on how to handle an 'absence of discussion'.

> Added qualifier: "Absence of discussion about climatic conditions may be indicative of regular rainfall, *especially if there are no or few reports of disruptions to phenomena that are regularly affected by rainfall extremes, such as harvests and travel*. Nevertheless, such an assumption necessarily comes with a degree of uncertainty." Reference made to Pfister 1995; Pfister et al., 2018. Lines: 146-9.

Lines 144-148 – this is a very long sentence – suggest you fragment.

> Agreed. Edit made.

Lines 155-157 – on a more pragmatic note, it is also very likely that they were interested in weather conditions as they relied upon them to grow their own food.

> This is implied on line 130 of the original submission (line 139 of the latest version), in which it is stated that droughts and floods could have an adverse impact "on Europeans' and surrounding societies' everyday lives...' Thus we've made no edit here.

Lines 169-178 – I'm slightly unclear over the methodology used here. Are you following directly the methodology used by Nicholson et al. (2012) whereby individual pieces of narrative evidence (i.e. individual quotes) are read and graded from 1-7 (and then averaged to give an annual picture), or the approach used in most other documentary-based climate reconstructions around the world where collections of quotes from specific months or seasons are read together and given a collective grade? This is important because, as Nash et al. (2021 – section 8.3) have discussed, the Nicholson method tends to lead to an over-representation of drier conditions in the resulting reconstruction. Figs 2-4 seem to suggest some sort of hybrid, which could be problematic if the authors are aiming to replicate Nicholson's approach as they suggest.

> We read and graded pieces of narrative evidence and used the archive to guide us on the time-period to which it referred. For example, a report from Tabora made in January stating that the rainy season had yet to begin would provide data for November-January. Alternatively, a similar report from March, would provide data for November-March. We've added sentences to clarify this. We are

not trying to replicate Nicholson et al 2012 – see above for explanation. Edits made on lines: 178-85.

Lines 199-203 – I would appreciate a little more explanation over the way in which individual diary entries are incorporated into Figs 2-4, particularly where they are merged with results from quotations in letters that could refer to conditions over periods of longer than a single day. In effect, you appear to be giving equal weight to (for example) a single daily diary entry describing drought and a letter documenting dry conditions that could span weeks or months. If you are following the Nicholson method described above this could lead to an over-representation of particular conditions, especially if these are isolated quotes from a personal diary rather than a 'weather diary' with daily weather-related entries.

Sentences added to clarify this, using the example of the one letter and 12 diary entries. See lines: 202-12.

Line 225 – there are many reasons for famine, not simply climatic. Do you have any contextual data from missionary sources that might explain the causes?

Yes. Added a sentence to reflect this with citations to missionary reports. We have also added a reference to Rockel 2022, which deals with this drought/famine in late-nineteenth century equatorial eastern Africa in depth. Lines: 247-53.

Lines 319-321 – this sounds like the approach used in the majority of historical climatology studies based on the European tradition, where monthly indices are summed and averaged. There is nothing new here methodologically, so you would do well to cite related sources – see section 8.1 in Nash et al. (2021) for more detail.

This is correct. Citations (Nash et al., 2021; Pfister et al., 2018) have been added. Lines: 352.

Lines 337-347 – I'm intrigued to know why you have adopted this approach when Nicholson has published her method for converting rain gauge data into 7-point index values based on standard deviations from the long-term mean (see section 8.3 in Nash et al. 2021 for a summary). If, as suggested earlier by the authors, they are trying to make results that are interoperable with those of Nicholson, then surely the same method needs to be used in this study?

It has now been clarified earlier in the paper that this study is not made to be exactly comparable or interoperable with Nicholson et al. 2012 – see above. We use our methodology (and not that of Nicholson et al. 2012) because it allows

conversion of the reanalysis and GCM data to the same scale as the documentary data, enabling direct comparisons and integration, whilst also maintaining the temporal resolution of both reanalysis and GCM datasets." (Lines 345-6 in first submission; lines 375-6 in revised submission).

Lines 513-517 – these sentences again oversimplify the apparent subjectivity of European observers. If you are going to make statements that European observers "regularly misunderstood the climatic and environmental contexts they reported on", then you need supporting evidence. I would suggest softening of these two sentences. There is as much evidence in the literature supporting the idea that European observers provided reliable eye-witness testimonies of climatic conditions in Africa as there is that their observations were unreliable.

> Revised to critique historians' interpretations of documentary materials more than the materials themselves: "A recurring theme is that such reports may often be more indicative of when droughts began to adversely affect the societies within which the Europeans lived or visited, not just when the drought set in. Thus, historical interpretations of, for example, historical famines, may be better understood as the result of an unfolding of longer-term climatic, environmental, and societal factors, rather than the onset of a sudden disaster, as permeates much of the historiography (cf. Rockel 2022; Matua and Runguma 2020)." Lines: 564-7.

Lines 522-525 – in light of the work by Mutua and Runguma (2020), this sentence requires revision.

> Agreed, change made: "Thus, direct evidence from the region in the form of historical documents, hitherto not incorporated into global climate reconstructions, allow the models to be deployed…" Lines: 575-7.

---

## Author Comment (AC2)

Many thanks for this paper. Whilst not an 'expert' in the field of documentary analysis, I approach this paper as a potential user of the outputs and the application of the methodology as a way for testing / comparing records derived from lake sediments against an independent archive. The approach using documentary evidence from Tanzania fill an important gap in knowledge in part of the world where historical (and palaeo!) records are still few and far between. I would also, before listing my corrections, like to apologise profusely to the authors for the delay in submitting this review. I have been in and out of the office for work, and as with everyone, workloads have simply got the better of me this year.

Overall, I think this is a good paper. I have some clarifications to elements of the text. And just being a little pedantic, the authors have over-used the number of figures; many of these should be panel figures or stacked figures. As they are all on the same timeframe, it would be so much easier to compare the figures generated from different datasets if they were stacked and became (for example) Figure 1 (a,b,c). I will also point out where this could be improved. Obviously, this is not a critical element to the robustness of the science presented, but to make it accessible and easier for the reader of the journal, I would strongly suggest this change to the presentation of figures is made.

Just an aside, I wondered whether you had worked with / engaged any Tanzanian scientists in this work, given this is the country of focus? Again, not critical to whether or not this paper is accepted, but the landscape is changing, and it would be wonderful to see Climate of the Past publishing work that is delivered in partnership with scientists from the country that is having science done to them.

> The authors acknowledge that the manuscript was prepared without collaboration with scholars in Tanzania. This is partly a result of all the documentary sources having been consulted for this article being available either in Europe or digitally. If our ideas and methodologies are to be taken forwards, however, then in-country research and collaboration will be a necessity. We alluded to this on lines 20, 95 and 541, in which we call for the integration of oral traditions for earlier periods. In the revised draft, we will be make a call for subsequent in-country research more explicit in the conclusion (line 541).

The remainder of clarifications are given by line number. Given I am not a technical expert, my comments are really directed to help users of your work (i.e., not historical / documentary analysts) to understand what you are presenting, and to provide them with data / figures that are easily comparable to other archives.

Ln25: I'm not sure I fully understand how the approach is complementary to / different from existing published methods? What issues will this cause of users of the approach used by Nicholson (for example)?

> We think the reviewer means to refer to around line 167. The point is well-taken, and we addressed a similar comment in our response to the first reviewer. We intend to obliviate the word 'interoperable' from this part of the manuscript, and instead we claim that our indexed time-series is 'broadly comparable' to that of Nicholson et al. (2012). We argue

that our method is more appropriate for integrating documentary evidence with outputs from reanalysis and GCMs.

Ln175: What is the cut-off used for high uncertainty vs low uncertainty? How did you determine this (i.e., had to score 3 out of 4 criteria)? How subjective is it? How transferable is this method from one researcher to another if it were developed, for example, in another country within Africa?

> We did not have a cut-off. In a revised draft, we will cite Adamson, Nash, and Grab (2022) (which we cite elsewhere) explicitly here. They argue that "variability between researchers should be considered minimal where index-based climate reconstructions are generated by trained historical climatologists working in groups of two or more." In short, the collaborative nature of our time-series creation has minimised individual subjectivities.

Ln187(ish): Figures 2-4. This really isn't a set of 3 separate figures. Even the combined heading that has been used suggest this. Rather, this is a stacked / panel diagram on a single page, on a single timescale labelled Figure 2a, b, c. Further, the text on this diagram needs to be larger / readable. It's a big strain to see what is there. Also, on this (and all diagrams with the timescale on the x-axis). What is the notation you use? Is 01 January, 02 February etc or is it seasonal? You need to make this clear somewhere (in figure caption), so we are all on the same page. Originally, I thought it was record 1 from 1855, record 02 etc. The way the notation is at the moment is just too hard to understand and read. Try to align all x-axis scales (you can do this if you change your chart-type – it will also save the pain of the axis labels being fixed).

> We will be happy to work with the editors of *Climate of the Past* to ensure that our figures are labelled and aligned optimally. In our original submission, we labelled them as 2.1, 2.2, 2.3 (rather than 2, 3, 4), but we were advised that they should be labelled only with integers. We will be happy to revert using a lettered system here instead, if that is preferred.

Ln187(ish): A follow up on the figure caption. No matter how many times I read it, I can't quite work out what you mean in the last line of the caption ('Values of +0.05…') you are going to need to spell this out in clearer language. I can't even work out where +0.05 would come from if your scale is -3 to +3 (or have I missed the point?). Can't you use a symbol or colour on the x-axis label to identify where there are no documentary data? There has to be a better way than arbitrarily adding a score. You may also want to increase the resolution of the image, my printer isn't wonderful, but the copy here is quite low res.

> We will rephrase the sentences on line 189 to: "The smallest colour bars/markings on the x-axis indicate an index value of 0. Absence of any markings indicates that there was no documentary data for that period."

Ln201: Somewhere here you start referring to 'documentary references for each line' – what's a line? Is this a bar on your graph? In which case you would refer to it by month and year (or season and year (see comment above). If this isn't what is meant here, this whole paragraph

needs a re-write to clarify exactly what you are referring to, because I can't quite follow t. Can you provide a supplementary data table that shows a; 'lines' and all data a source used in the paper. At the moment there is no transparency, just your interpretation here. Providing the underpinning data would help the reader better understand the approach you are taking, especially when the text 9as written) is quite complex to understand.

This is a good point, and one which is similar to one of the first reviewer's (their comment refers to lines 199-203). We've added several sentences to explain what comprises a line of data, by using the line with 12 references as a reference. The full paragraph will now read:

"The three time-series reflect a total of 151 lines of data: 55 for Mpwapwa; 52 for Tabora; and 44 for Ujiji. The datasets underpinning figures 2–4, including transcriptions and comments on individual references, are available by link in the section, 'Data Availability'. Each line of data refers to climate conditions during a given period, ranging from one month to an entire rainy season. The number of documentary references for each line is between one (several) and twelve (line 31 of the Ujiji dataset). The latter example refers to November–December 1886, and includes data from a letter written by an LMS missionary based at Kavala Island and eleven separate diary entries by a White Fathers' missionary based at Kibanga. At Kavala Island, the LMS representative reported in early January that farmers had recently gone to the mainland to cultivate, which is later than usual and could suggest a slightly delayed beginning to the rainy season (CWM/LMS/06/02/012 Lea to Thompson, 6 Jan 1887). However, the evidence from Kibanga has much more granular data, with reports of 'clouds… gathering' and abundant rain from the end of October (A.G.M.Afr. Diaire de Kibanga, 19 Oct. 1886, 22 Oct 1886, 25 Oct. 1886, 27 Oct. 1886, 10 Nov. 1886, 22 Nov. 1886; 26 Nov. 1886, 30 Nov. 1886). That these were not just episodes of rainfall, but part of a broader trend is confirmed by reports written in December, during which heavy rainfall caused a nearby stream to overflow on 1 December, following which the missionaries were able to plant wheat on 6 December, and on 16 December there were 'big rains… these days' (A.G.M.Afr. Diaire de Kibanga, 1 Dec. 1886, 6 Dec. 1886, 16 Dec. 1886). Read collectively, it was determined that these reports suggested slightly above average rainfall in Ujiji (index value = 1) for November–December 1886 (cf. Nash et al., 2021). However, as the reports were made outside rather than within Ujiji, this datapoint only received a confidence value of 1. In addition to published data by 'explorers,' which provide references for all locales, data for Mpwapwa is informed principally by documents held in the CMS archive, with documents from the White Fathers' archive providing additional information for 1880-82; the Tabora dataset is equally informed by documents in the CMS and White Fathers' archives, with occasional references to the LMS archive; and the Ujiji dataset is informed equally by references in the LMS and White Fathers' archives, as well as by one reference from the AIA archive.

Note, we have provided the underpinning data in the 'Data Availability' section, which we've referred to directly in the second sentence of the paragraph. Given that this is freely available, we don't see the need to include table with some of the underpinning data in-text.

Ln210: How do you define the 'quality' of the data? What were the parameters that you set? They would need to be presented here (or in a supplementary document) also.

> 'Quality' was used in this instance as a synonym for confidence. On reflection, this was an error. Thus, we will change the sentence to: "As expected, there is limited quantity of data, especially data for which we attribute high levels of confidence, for the first twenty years…"

Ln280: As previously, these are not 6 different figures. It is a single figure with multiple panels. They all need to be stacked and appear on a single page to enable the reader to make an assessment of the data you are providing. They also need to be labelled sequentially (i.e., not refer to 5, 7,9, but to 5,6,7 – or my preference would be a,b,c). Perhaps consider a different way to display these – there are lots of examples in the literature that would help you to better present these. At the minute it's all very confusing and I don't think will land with the audience of CoP. At a push this could be 2 figures 5a,b,c and 6a,b,c where you would have a column for each model. They are related (a-c), not independent (5-10) figures. Follow the layout for your Figure 11 – you have not labelled these as 25 different figures!

> As with the earlier comment, we are happy to work with the editors to label and align the figures appropriately.

Ln312: I know it's going to be hard, but the text size is quite small in your figure 11, is there an outside chance you can make it slightly bigger?

> On reflection, this is the least important of all the figures to have in-text. Instead of making each graph bigger for clarity, which would necessitate it passing over more than one page, we suggest – if it is acceptable to the editors – including this figure as a 'supplementary figure,' with a link to it at the end of the article.

Ln327: Another figure. You could approach it that in any given panel figure 'a' is always Mpwapwa, 'b' is always Tabora, and 'c' is Ujiji. That will also make it easier for the reader to follow the story. You need to take the reader along with you, not make it overly complex for them to engage.

> As with the earlier comment, we are happy to work with the editors to label and align the figures appropriately.

Ln360: Again, think of the labelling. If you do want to keep these as separate figures, they would all need separate captions (i.e. one for Fig 15, a new one for Fig 16 and another for Fig 17; they can't be all as one. In doing so, you are suggesting to me they are related and should be presented slightly differently.

> As with the earlier comment, we are happy to work with the editors to label and align the figures appropriately.

---

## Author Response (AR1)

**Reply to Comments**

In this document, we explain the changes that we made in the creation of our revised manuscript. Line numbers in the responses refer to those of the clean revised version. Once again, we would like to thank the reviewers for their kind and considered comments.

**RC1**

This is a welcome paper addressing the thorny issue of how best to integrate historical and meteorological evidence to reconstruct climates of the past. It is clearly structured, generally very well written and makes persuasive arguments. However, as you will see from my specific comments below, it includes some statements that are problematic. These mainly relate to the way that the text falls into the "documentary evidence unreliable, instrumental data reliable" trope common to many climatological studies.

> Agreed. We have made several edits to revise the tone of the work so that the documentary evidence is given more of its due. Our intention was certainly not to diminish the utility and importance of the documentary-derived data – our time-series rely significantly on it, after all. Notable edits are:
>
> Lines 52-3: … despite occasionally including reports on weather and weather-affected phenomena that can **regularly be transformed into time-series of climate variability (cf. Mutua and Runguma, 2020)**…
>
> Lines 123-5: Although each document-type provides somewhat distinct **opportunities** and challenges for creating an indexed time-series of rainfall, there are some prevailing themes. **In general, 'explorers' and missionaries were highly interested in documenting climatic conditions and variations**, although their interest varied over time and space, and between each author.
>
> Lines 182-4: The decision to use a 7-point system instead of a 5-point or 3-point system, which are also commonly used in climate indices, reflects the **high level of granularity in many of the documentary reports, especially from missionaries** (cf. Pfister et al., 2018).

There are also some methodological issues that need addressing. These include the need for a clearer description of the way in which documentary index classes are derived.

> We've added descriptions to Table 1 (lines 199-203) which explain how different features of the documentary source material were categorized into index classes.

Explanation is also needed as to why – given that the goal of the study is to produce a time series that is "interoperable with Nicholson et al. (2012)" (line 167) – the method used to convert modelled rainfall levels from 20CR and GCMs is not the same as that used by Nicholson et al. (2012).

> We've modified the expressed goal of the study. It is not to make it interoperable with Nicholson et al. (2012). Rather, in deploying a 7-point scale, which reflects the high level of

granularity in some of the documentary data (lines 182-4), we have produced outputs that are broadly comparable to Nicholson et al. (2012), as they, too, used a 7-point scale, although they used different methods. Our in-text explanation is: "This makes its outputs broadly comparable to those from Nicholson, Dezfuli, and Klotter's 2012 dataset for Africa's nineteenth-century rainfall, which also uses a seven-point system, even if the different methods and sources used mean that the datasets are not entirely interoperable" (lines 180-2).

I'm very surprised that the authors do not cite the recent papers by Nash et al. (2018) and Mutua & Runguma (2020), which present 19[th] century documentary climate series for Malawi and Kenya respectively. I would have thought that these are essential for comparison with the results presented for Tanzania.

> We've added several references to these articles which help to situate our interpretations in a wider geographical focus. Further, we've cited rain gauge data captured at stations in Kampala, Mombasa, and Zanzibar for the same purpose. Explanation for this approach is on lines 93-95, as well as in the abstract (line 20).

Lines 16-17, 60 and 507 – I'm not sure the wording "...a more scientifically grounded interpretation of documentary materials..." in the abstract and main text is ideal. This makes a value judgement about the validity of documentary evidence. Maybe 'climatologically grounded' rather than 'more scientifically' grounded would be better.

> This is a good point. We have changed it to, 'an interpretation of documentary materials that is grounded in both the humanities and natural sciences.' See lines: 16-17, 64-5, 570-1.

Line 38 (and throughout) – best to avoid the use of terms derived from 'East Africa' as this is a colonial construct. Academics that I have worked with from the region tend to prefer the term 'eastern Africa'.

> This is true, and something Philip Gooding (Author 1) addressed in his recent monograph (*On the Frontiers of the Indian Ocean World: A History of Lake Tanganyika, c.1830-1890*, Cambridge, 2022, p. xii). Unfortunately, 'eastern Africa' doesn't really work either, as the term can be (and has been) applied to anywhere in the eastern half of the continent, from Cairo in the North to the Cape in the South. Thus, it lacks specificity. The better term is 'equatorial eastern Africa,' but this is perhaps overly long and its usage may hinder readability. We also noted that several climatological studies (e.g. Alin and Cohen 2003; Bessems et al., 2008; Hastenrath 2001; Nicholson 2015; Nicholson and Yin 2001; Verschuren et al. 2000) use 'East Africa', and so we thought we were conforming to established practice. But, in the interest of challenging colonial constructs, perhaps this practice needs to change. **Thus, we have changed all references to East Africa to equatorial eastern Africa.** However, if there are objections related to readability, we would rather revert to East Africa (which is usually understood to refer to Tanzania, Kenya, Uganda, Burundi, Rwanda, and the eastern DRC) than to adopt 'eastern Africa,' because of the latter's lack of specificity.

Line 45 – I don't think Endfield & Nash (2002) used a term as strong as 'distorted' to describe European perceptions (and hence descriptions) of African climate. Rather, the descriptions made by Europeans were often framed relative to their 'home' climate (particularly during their early years of residence in Africa), so tend to over-emphasise drier conditions. As noted in section 2, they may also be shaped by imperial knowledge-making.

> We've clarified this. True – their understandings of climate on its own were not necessarily 'distorted.' However, the role that they saw climate playing in the regions they reported certainly was i.e. they conceptualisation was rooted in enviro-climatic determinism. We've made an edit to reflect this and taken out the reference to Endfield and Nash 2002, which does not deal with this issue as much as the other cited texts. See lines: 46-8: "They were written almost entirely by Europeans, including by so-called 'explorers' and missionaries, who regularly understood the peoples, environments, and climates they encountered through frameworks underpinned by enviro-climatic determinism (Gooding, 2019; Gooding, 2022a; Rockel, 2022)."

Lines 46-48 – this is a very strong statement – are you sure that absolutely no records made by Europeans describing climate in Tanzania exist between 1861 and 1868? I find that very hard to believe.

> This is actually correct, although we've made an edit to clarify. No Europeans entered the region between Speke/Grant's expedition (which ended in 1861) and David Livingstone's arrival in Ujiji in 1869. The clarification we've made is to point out that there were no first-hand reports between these dates (this is similar to inland Kenya, as Matua and Runguma (2020) point out as well). There may have been the odd second-hand report made by administrators in Zanzibar, who may have consulted with Omani/African traders and/or porters who had returned from inland regions, but we have not made an effort to find them, if they even exist. If they do, they will not provide the kind of granular data that the firsthand reports that we have consulted contain. Lines 49-50: "For example, although the first Europeans to document enviro-climatic conditions first-hand in inland Tanzania did so in 1856–61, another did not do so again until 1869 (Burton, 1860; Speke, 1864a; Speke, 1864b; Livingstone, 1875)."

Lines 114-127 – this paragraph makes some valuable points. However, it paints the rather sweeping picture that all explorer and missionary descriptions of weather and climate were shaped by imperial agendas and are therefore unreliable. Some descriptions might well be 'highly subjective' – especially the broad overviews of climatic conditions in the more general explorer monographs – but other accounts of specific weather events and related phenomena (e.g. delays to the start of the rainy season, counts of rainy days, descriptions of flood events, descriptions of pasture conditions etc) will likely be reliable. I suggest that this paragraph be tweaked to provide greater nuance.

> Yes, the paragraph, as it stood, skewed overly negative. Thus, we have added 'positive' aspects of the documentary material as well: "Descriptions of rainy days, flood events, pasture conditions, and harvests, which are all to varying degrees related to climatic conditions, are abundant and can be used to make time-series of rainfall variability (Nash et al., 2018; Mutua and Runguma 2020)" (Lines 125-7). We've kept the references to imperial knowledge-making, however. We feel it important to acknowledge how and why

these documents exist, as well their utility to scholars in the present (the latter being much more well-known).

Lines 128-129 – the emphasis on describing extreme conditions is not unique to African documentary evidence and is well documented in historical climatology studies around the world – have a look at some of the excellent reviews by Christian Pfister or Rudolf Brazdil for further details and cite relevant methodological sources.

> Thank you. Added references to Pfister 1995, Brazdil 2000, and Brazdil et al. 2005, as well as a call out to this wider research in the opening line of the paragraph: "Given this historical background (and in line with other documentary sources for other regions and time periods), it is probably unsurprising that Europeans commented on climatic conditions more when they were extreme, such as in instances of severe drought or floods, than during months/seasons/years of regular rainfall" (Lines 137-40).

Lines 130-139 – these kinds of uncertainty surrounding 'climatically indirect' indicators of climate variability are routinely dealt with in historical climatology studies and there is a wide literature on this. Again, have a look at some of the reviews by Christian Pfister or Rudolf Brazdil for details. These include explicit guidance on how to handle an 'absence of discussion'.

> Added qualifier: "Absence of discussion about climatic conditions may be indicative of regular rainfall, **especially if there are no or few reports of disruptions to phenomena that are regularly affected by rainfall extremes, such as harvests and travel.** Nevertheless, such an assumption necessarily comes with a degree of uncertainty (cf. Pfister 1995; Pfister et al., 2018)" (Lines 147-51).

Lines 144-148 – this is a very long sentence – suggest you fragment.

> Agreed. Edit made, lines 156-160: The first is that the rainy season was particularly long in 1857–8, and that he made assumptions based on his own experience. The alternative is that he mistakenly integrated secondary information about the 'lake regions' of eastern Africa into his assessment of Ujiji's climate: a September–May rainy season broadly aligns with conditions on the northern shores of Lake Victoria, even if December–February is usually drier than September–November and March–May in the latter place.

Lines 155-157 – on a more pragmatic note, it is also very likely that they were interested in weather conditions as they relied upon them to grow their own food.

> This is implied on line 130 of the original submission (lines 140-1 of the revised ms), in which it is stated that droughts and floods could have an adverse impact "on Europeans' and surrounding societies' everyday lives…' Thus, we've made no edit here. ("Droughts and floods had adverse impacts on Europeans' and surrounding societies' everyday lives, and so were deemed worthy of reporting".)

Lines 169-178 – I'm slightly unclear over the methodology used here. Are you following directly the methodology used by Nicholson et al. (2012) whereby individual pieces of narrative evidence (i.e. individual quotes) are read and graded from 1-7 (and then averaged to give an annual picture), or

the approach used in most other documentary-based climate reconstructions around the world where collections of quotes from specific months or seasons are read together and given a collective grade? This is important because, as Nash et al. (2021 – section 8.3) have discussed, the Nicholson method tends to lead to an over-representation of drier conditions in the resulting reconstruction. Figs 2-4 seem to suggest some sort of hybrid, which could be problematic if the authors are aiming to replicate Nicholson's approach as they suggest.

> We read and graded pieces of narrative evidence and used the archive to guide us on the time-period to which it referred. For example, a report from Tabora made in January stating that the rainy season had yet to begin would provide data for November-January. Alternatively, a similar report from March, would provide data for November-March. We've added sentences to clarify this. We are not trying to replicate Nicholson et al 2012 – see above for explanation (response to query on line 167 [original version]. Edits made on lines [new version] 179-86: "Notwithstanding these challenges, this article uses a 7-point index system to quantify the qualitative descriptions of rainfall variability and its effects. This makes its outputs broadly comparable to those from Nicholson, Dezfuli, and Klotter's 2012 dataset for Africa's nineteenth-century rainfall, which also uses a seven-point system, even if the different methods and sources used mean that the datasets are not entirely interoperable. The decision to use a 7-point system instead of a 5-point or 3-point system, which are also commonly used in climate indices, reflects the high level of granularity in many of the documentary reports, especially from missionaries (cf. Pfister et al., 2018). The definitions of each index value are displayed in table 1, along with descriptions about how the documentary index classes were derived from qualitative descriptions. Narrative evidence was collaboratively read and then graded for the time-period to which the document(s) referred according to these definitions (cf. Adamson et al., 2022)."

Lines 199-203 – I would appreciate a little more explanation over the way in which individual diary entries are incorporated into Figs 2-4, particularly where they are merged with results from quotations in letters that could refer to conditions over periods of longer than a single day. In effect, you appear to be giving equal weight to (for example) a single daily diary entry describing drought and a letter documenting dry conditions that could span weeks or months. If you are following the Nicholson method described above this could lead to an over-representation of particular conditions, especially if these are isolated quotes from a personal diary rather than a 'weather diary' with daily weather-related entries.

> Sentences added to clarify this, using the example of the one letter and 12 diary entries. See lines 219-32: Each line of data refers to climate conditions during a given period, ranging from one month to an entire rainy season. The number of documentary references for each line is between one (several) and twelve (line 31 of the Ujiji dataset). The latter example refers to November–December 1886, and includes data from a letter written by an LMS missionary based at Kavala Island and eleven separate diary entries by a White Fathers' missionary based at Kibanga. At Kavala Island, the LMS representative reported in early January that farmers had recently gone to the mainland to cultivate, which is later than usual and could suggest a slightly delayed beginning to the rainy season (CWM/LMS/06/02/012 Lea to Thompson, 6 Jan 1887). However, the evidence from Kibanga has much more granular data, with reports of 'clouds... gathering' and abundant rain from the end of October (A.G.M.Afr. Diaire de Kibanga, 19 Oct. 1886, 22 Oct 1886, 25 Oct. 1886, 27 Oct. 1886, 10 Nov. 1886, 22 Nov. 1886; 26 Nov. 1886, 30 Nov. 1886). That these were not

just episodes of rainfall, but part of a broader trend is confirmed by reports written in December, during which heavy rainfall caused a nearby stream to overflow on 1 December, following which the missionaries were able to plant wheat on 6 December, and on 16 December there were 'big rains... these days' (A.G.M.Afr. Diaire de Kibanga, 1 Dec. 1886, 6 Dec. 1886, 16 Dec. 1886). Read collectively, it was determined that these reports suggested slightly above average rainfall in Ujiji (index value = 1) for November–December 1886 (cf. Nash et al., 2021).

Line 225 – there are many reasons for famine, not simply climatic. Do you have any contextual data from missionary sources that might explain the causes?

Yes. Added a sentence to reflect this with citations to missionary reports. We have also added a reference to Rockel 2022, which deals with this drought/famine in late-nineteenth century equatorial eastern Africa in depth. Lines: 254-60: This scenario at from Mpwapwa is made furthermore uncertain because the subsequent season coincided with widespread famine. This may be indicative of previous harvests being insufficient, triggered by deficient levels of rainfall, decreasing societal resilience to region-wide drought in 1883–4, and so putting into question the validity of observations gathered from missionary correspondence in 1882–3 (Rockel 2022). In any case, missionary reports are unequivocal that the underlying cause of famine in 1883–4 was drought, even if structural factors may additionally have exacerbated shortages (cf. CMS G/3/A/6/O Price to Lang, 2 May 1884; CMS G/3/A/6/O Price to Lang, 2 May 1884; CMS G/3/A/6/O Price to Lang 5 Aug. 1884).

Lines 319-321 – this sounds like the approach used in the majority of historical climatology studies based on the European tradition, where monthly indices are summed and averaged. There is nothing new here methodologically, so you would do well to cite related sources – see section 8.1 in Nash et al. (2021) for more detail.

This is correct. Citations (Nash et al., 2021; Pfister et al., 2018) have been added. Line: 361.

Lines 337-347 – I'm intrigued to know why you have adopted this approach when Nicholson has published her method for converting rain gauge data into 7-point index values based on standard deviations from the long-term mean (see section 8.3 in Nash et al. 2021 for a summary). If, as suggested earlier by the authors, they are trying to make results that are interoperable with those of Nicholson, then surely the same method needs to be used in this study?

It has now been clarified earlier in the paper that this study is not made to be exactly interoperable with Nicholson et al. 2012 – see above. We use our methodology (and not that of Nicholson et al. 2012) because it allows conversion of the reanalysis and GCM data to the same scale as the documentary data, enabling "direct comparisons and integration, whilst also maintaining the temporal resolution of both reanalysis and GCM datasets" (Lines 388-9).

Lines 513-517 – these sentences again oversimplify the apparent subjectivity of European observers. If you are going to make statements that European observers "regularly misunderstood the climatic and environmental contexts they reported on", then you need supporting evidence. I would suggest softening of these two sentences. There is as much evidence in the literature

supporting the idea that European observers provided reliable eye-witness testimonies of climatic conditions in Africa as there is that their observations were unreliable.

> Revised to critique historians' interpretations of documentary materials more than the materials themselves: "A recurring theme is that such reports may often be more indicative of when droughts began to adversely affect the societies within which the Europeans lived or visited, not when the drought set in. Thus, historical interpretations of, for example, past famines in the region, may be better understood as the result of an unfolding of longer-term climatic, environmental, and societal factors, rather than the onset of a sudden disaster, as permeates some of the historiography (cf. Rockel, 2022; Mutua and Runguma, 2020; Gooding 2023)" (Lines: 574-8).

Lines 522-525 – in light of the work by Mutua and Runguma (2020), this sentence requires revision.

> Agreed, change made: "Thus, direct evidence from the region in the form of historical documents, hitherto not incorporated into global climate reconstructions, allow the models to be deployed..." Lines: 586-7.

**RC2**

Just an aside, I wondered whether you had worked with / engaged any Tanzanian scientists in this work, given this is the country of focus? Again, not critical to whether or not this paper is accepted, but the landscape is changing, and it would be wonderful to see Climate of the Past publishing work that is delivered in partnership with scientists from the country that is having science done to them.

> The authors acknowledge that the manuscript was prepared without collaboration with scholars in Tanzania. This is partly a result of all the documentary sources having been consulted for this article being available either in Europe or digitally. If our ideas and methodologies are to be taken forwards, however, then in-country research and collaboration will be a necessity. We alluded to this on lines 20, 95 and 541, in which we call for the integration of oral traditions for earlier periods. In the revised draft, we have made the call for subsequent in-country research and collaboration more explicit, especially in the conclusion: "In-region research and collaboration will be necessary for such research to be undertaken" (Lines 605-6).

Ln25: I'm not sure I fully understand how the approach is complementary to / different from existing published methods? What issues will this cause of users of the approach used by Nicholson (for example)?

> We think the reviewer means to refer to around line 167. The point is well-taken, and we addressed a similar comment in our response to the first reviewer. We have obliviated the word 'interoperable' from this part of the manuscript, and instead we claim that our indexed time-series is 'broadly comparable' to that of Nicholson et al. (2012). We argue

that our method is more appropriate for integrating documentary evidence with outputs from reanalysis and GCMs.

Lines 180: "This makes its outputs **broadly comparable** to those from Nicholson, Dezfuli, and Klotter's 2012 dataset…"

Lines 387-9: "This approach therefore allowed conversion of the reanalysis and GCM data to the same scale as the documentary data, enabling direct comparisons and integration."

Ln175: What is the cut-off used for high uncertainty vs low uncertainty? How did you determine this (i.e., had to score 3 out of 4 criteria)? How subjective is it? How transferable is this method from one researcher to another if it were developed, for example, in another country within Africa?

We did not have a cut-off. We have now cited Adamson, Nash, and Grab (2022) (which we had already cited elsewhere) explicitly here. They argue that "variability between researchers should be considered minimal where index-based climate reconstructions are generated by trained historical climatologists working in groups of two or more." In short, the collaborative nature of our time-series creation has minimised individual subjectivities. We also edited the sentence to emphasise the collaborative nature of the process: "Narrative evidence was collaboratively read and then graded for the time-period to which the document(s) referred according to these definitions (cf. Adamson et al., 2022) (line 186-7).

Ln187(ish): Figures 2-4. This really isn't a set of 3 separate figures. Even the combined heading that has been used suggest this. Rather, this is a stacked / panel diagram on a single page, on a single timescale labelled Figure 2a, b, c. Further, the text on this diagram needs to be larger / readable. It's a big strain to see what is there. Also, on this (and all diagrams with the timescale on the x-axis). What is the notation you use? Is 01 January, 02 February etc or is it seasonal? You need to make this clear somewhere (in figure caption), so we are all on the same page. Originally, I thought it was record 1 from 1855, record 02 etc. The way the notation is at the moment is just too hard to understand and read. Try to align all x-axis scales (you can do this if you change your chart-type – it will also save the pain of the axis labels being fixed).

Ln327: Another figure. You could approach it that in any given panel figure 'a' is always Mpwapwa, 'b' is always Tabora, and 'c' is Ujiji. That will also make it easier for the reader to follow the story. You need to take the reader along with you, not make it overly complex for them to engage.

Ln280: As previously, these are not 6 different figures. It is a single figure with multiple panels. They all need to be stacked and appear on a single page to enable the reader to make an assessment of the data you are providing. They also need to be labelled sequentially (i.e., not refer to 5, 7,9, but to 5,6,7 – or my preference would be a,b,c). Perhaps consider a different way to display these – there are lots of examples in the literature that would help you to better present these. At the minute it's all very confusing and I don't think will land with the audience of CoP. At a push this could be 2 figures 5a,b,c and 6a,b,c where you would have a column for each model. They are related (a-c), not independent (5-10) figures. Follow the layout for your Figure 11 – you have not labelled these as 25 different figures!

Ln360: Again, think of the labelling. If you do want to keep these as separate figures, they would all need separate captions (i.e. one for Fig 15, a new one for Fig 16 and another for Fig 17; they can't be all as one. In doing so, you are suggesting to me they are related and should be presented slightly differently.

> We have yet to make changes in response to any of these four comments. In our original submission, we labelled them more similarly to how the reviewer envisaged e.g. 2.1, 2.2, and 2.3 instead of 2, 3, and 4. However, we changed them at the request of the *Climate of the Past* editors. We will gladly work with the editors to optimise labelling moving forwards.

Ln201: Somewhere here you start referring to 'documentary references for each line' – what's a line? Is this a bar on your graph? In which case you would refer to it by month and year (or season and year (see comment above). If this isn't what is meant here, this whole paragraph needs a re-write to clarify exactly what you are referring to, because I can't quite follow t. Can you provide a supplementary data table that shows a; 'lines' and all data a source used in the paper. At the moment there is no transparency, just your interpretation here. Providing the underpinning data would help the reader better understand the approach you are taking, especially when the text 9as written) is quite complex to understand.

> This is a good point, and one which is similar to one of the first reviewer's (their comment refers to lines 199-203). We've added several sentences to explain what comprises a line of data, by using the line with 12 references as a reference. The full paragraph now reads:
>
> "The three time-series reflect a total of 151 lines of data: 55 for Mpwapwa; 52 for Tabora; and 44 for Ujiji. The datasets underpinning figures 2–4, including transcriptions and comments on individual references, are available by link in the section, 'Data Availability'. Each line of data refers to climate conditions during a given period, ranging from one month to an entire rainy season. The number of documentary references for each line is between one (several) and twelve (line 31 of the Ujiji dataset). The latter example refers to November–December 1886, and includes data from a letter written by an LMS missionary based at Kavala Island and eleven separate diary entries by a White Fathers' missionary based at Kibanga. At Kavala Island, the LMS representative reported in early January that farmers had recently gone to the mainland to cultivate, which is later than usual and could suggest a slightly delayed beginning to the rainy season (CWM/LMS/06/02/012 Lea to Thompson, 6 Jan 1887). However, the evidence from Kibanga has much more granular data, with reports of 'clouds... gathering' and abundant rain from the end of October (A.G.M.Afr. Diaire de Kibanga, 19 Oct. 1886, 22 Oct 1886, 25 Oct. 1886, 27 Oct. 1886, 10 Nov. 1886, 22 Nov. 1886; 26 Nov. 1886, 30 Nov. 1886). That these were not just episodes of rainfall, but part of a broader trend is confirmed by reports written in December, during which heavy rainfall caused a nearby stream to overflow on 1 December, following which the missionaries were able to plant wheat on 6 December, and on 16 December there were 'big rains... these days' (A.G.M.Afr. Diaire de Kibanga, 1 Dec. 1886, 6 Dec. 1886, 16 Dec. 1886). Read collectively, it was determined that these reports suggested slightly above average rainfall in Ujiji (index value = 1) for November–December 1886 (cf. Nash et al., 2021). However, as the reports were made outside rather than within Ujiji, this datapoint only received a confidence value of 1. In addition to published data by 'explorers,' which provide references for all locales, data for Mpwapwa is informed principally by documents held in the CMS archive, with documents from the White Fathers' archive providing additional

information for 1880-82; the Tabora dataset is equally informed by documents in the CMS and White Fathers' archives, with occasional references to the LMS archive; and the Ujiji dataset is informed equally by references in the LMS and White Fathers' archives, as well as by one reference from the AIA archive" (Lines 217-37)

Note, we have provided the underpinning data in the 'Data Availability' section, which we've referred to directly in the second sentence of the paragraph. Given that this is freely available, we don't see the need to include table with some of the underpinning data in-text.

Ln210: How do you define the 'quality' of the data? What were the parameters that you set? They would need to be presented here (or in a supplementary document) also.

'Quality' was used in this instance as a synonym for confidence. On reflection, this was an error. Thus, we have changed the sentence to: "As expected, there is limited quantity of data, especially data for which we attribute high levels of confidence, for the first twenty years…" (lines 238-9).

Ln312: I know it's going to be hard, but the text size is quite small in your figure 11, is there an outside chance you can make it slightly bigger?

On reflection, this is the least important of all the figures to have in-text. Instead of making each graph bigger for clarity, which would necessitate it passing over more than one page, we have – if it is acceptable to the editors – included this figure as a supplementary figure.